# Proximity does not contribute to activity enhancement in the glucose oxidase–horseradish peroxidase cascade

Yifei Zhang[1], Stanislav Tsitkov[1] & Henry Hess[1]

A proximity effect has been invoked to explain the enhanced activity of enzyme cascades on DNA scaffolds. Using the cascade reaction carried out by glucose oxidase and horseradish peroxidase as a model system, here we study the kinetics of the cascade reaction when the enzymes are free in solution, when they are conjugated to each other and when a competing enzyme is present. No proximity effect is found, which is in agreement with models predicting that the rapidly diffusing hydrogen peroxide intermediate is well mixed. We suggest that the reason for the activity enhancement of enzymes localized by DNA scaffolds is that the pH near the surface of the negatively charged DNA nanostructures is lower than that in the bulk solution, creating a more optimal pH environment for the anchored enzymes. Our findings challenge the notion of a proximity effect and provide new insights into the role of DNA scaffolds.

[1] Department of Biomedical Engineering, Columbia University, New York, New York 10027, USA. Correspondence and requests for materials should be addressed to H.H. (email: hh2374@columbia.edu).

Enzyme cascades, which sequentially perform multiple enzymatic reactions, have a critical role in signal transduction and amplification in biological systems and also have a wide range of potential applications in biotechnology[1–5]. In the past decade, researchers developed strategies to construct artificial multienzyme systems[6,7]. In particular, the rapid development of DNA nanotechnology provides a programmable tool for the spatial organization of enzymes at specific sites[8]. When the components of an enzyme cascade were placed within nanometers on a DNA scaffold, a several-fold enhancement of the cascade throughput was observed[8–12]. These observations attracted a strong and growing interest in multienzyme catalysis, but the mechanism of the activity enhancement is still not clear[13].

Glucose oxidase (GOx) and horseradish peroxidase (HRP) are the most frequently used enzyme pair to demonstrate the feasibility and advantages of multienzyme systems on DNA scaffolds. Müller and Niemeyer[14] presented the earliest example of a GOx–HRP complex by nucleic acid hybridization in 2008. Wilner et al.[10] tethered GOx and HRP molecules to a DNA ribbon with a precisely defined distance between GOx and HRP pairs. By taking advantage of DNA origami, Fu et al.[11] created an individual GOx and HRP pair on each DNA scaffold. A several-fold enhancement of the cascade throughput relative to the throughput obtained from equal numbers of free GOx and HRP molecules in solution was always observed, and a closer placement gave a higher overall activity. Several other enzyme cascades also showed enhanced activity when they are co-assembled with DNA nanostructures[12,15,16]. The activity enhancement was frequently attributed to facilitated transport, or so-called substrate channeling, where a shorter distance between the coupled enzymes leads to a faster transfer of a newly produced intermediate substrate molecule to the second enzyme.

However, the mechanistic understanding of these experiments is still unsatisfactory. First, although the enzyme pairs were constructed by similar strategies, the activity enhancements vary a lot. For instance, Fu et al.[11] co-assembled the GOx–HRP pair on DNA origami and observed 15 times higher activity when the distance between two enzymes is 10 nm. In contrast, Xin et al.[17] placed the GOx and HRP even closer using a similar strategy but found only a 1.5-fold activity enhancement. Second, a theoretical analysis of the GOx–HRP cascade by Idan and Hess[18,19] challenged the proximity channeling effect. They estimated the timescale when the intermediate flux from the upstream enzyme is equal to the flux from the bulk solution and pointed out that the facilitated transport owing to proximity is a temporary effect that only enhances cascade throughput until the concentration of intermediate substrate molecules in the solution becomes significant. The duration of this normally very short-lived effect (on the millisecond timescale) can be extended by aggregating the enzyme pairs and/or introducing attractive interactions between the intermediate substrate and the scaffold. In brief, the aggregation provides more target enzymes in the vicinity of the first enzyme, and the attractive interaction can increase the local concentration of intermediate and prevent its leaking to the bulk solution. Ultimately, the overall activity will be limited by the maximal reaction rate of the slower enzyme[13,20]. Proximity in itself should not influence the maximal reaction rate of either enzyme; however, the reported activity enhancements appear to persist over the entire observation time which would indicate that the limiting maximal reaction rate has increased[11,17,21]. This contradiction between the theoretical analysis and experimental observations has not yet been resolved.

Here we examined the putative proximity effect for the GOx–HRP cascade without relying on DNA nanotechnology.

We place GOx and HRP within 2 nm of each other using a crosslinker, show that the catalytic activity of both enzymes is retained and demonstrate that there is no enhancement in cascade throughput (either transiently or permanently) and no direct substrate transfer. As the proximity channeling is ruled out, the DNA scaffold must be responsible for the previously observed activity enhancement. We measure the pH dependence of the maximal reaction rate of GOx and HRP and find that the observed activity enhancement can be explained by the lower pH near the DNA surface where the enzyme is located. These findings challenge the traditional view of proximity channeling and provide new insights into the role of DNA scaffolds.

## Results

**Kinetics of GOx and HRP.** GOx is a bi-substrate enzyme that oxidizes β-D-glucose with oxygen and produces glucono-δ-lactone and hydrogen peroxide ($H_2O_2$; Fig. 1a). First we showed that the oxygen concentration is saturating for the duration of the experiment (1 h) by measuring the oxygen concentration directly with an oxygen meter in a sealed vessel containing 1 nM GOx in 1 mM glucose (Fig. 1b). The oxygen consumption rate is constant at $15\,nM\,s^{-1}$ demonstrating that the reaction is not slowed by oxygen depletion or product inhibition. An HRP-coupled colorimetric assay quantified the rate of formation of $ABTS^{+\bullet}$ in the presence of 20 nM HRP and 2 mM 2,2′-azino-bis(3-ethylbenzothiazoline-6-sulfonic acid)-diammonium salt (ABTS). Consumption of 54 μM oxygen in 1 h corresponded to the formation of 112 μM $ABTS^{+\bullet}$, proving that the stoichiometry of $H_2O_2$ to $ABTS^{+\bullet}$ is 1:2. It also confirmed that the conventional assay using the GOx/HRP cascade is reliable as the high concentration of HRP (20 nM) ensures the rapid and complete conversion of $H_2O_2$ produced by 1 nM GOx. The colorimetric assay yielded a rate of $H_2O_2$ generation of $15.4 \pm 0.1\,nM\,s^{-1}$ (mean ± s.d.) in 1 mM glucose (the Michaelis–Menten kinetics of glucose is shown in Supplementary Fig. 1). The kinetics of HRP was determined by adding variable concentrations of $H_2O_2$ to 1 nM HRP and 2 mM ABTS (Fig. 1c). HRP follows Michaelis–Menten kinetics with a $K_m$ of HRP for $H_2O_2$ of $2.55 \pm 0.11\,\mu M$ and a turnover number $k_{cat}$ of $32.7 \pm 0.4\,s^{-1}$ (mean ± s.e.). The concentration of the $H_2O_2$ stock solution has been determined by measuring the amount of $ABTS^{+\bullet}$, which can be generated from a given $H_2O_2$ solution by HRP (Supplementary Figs 2 and 3).

**Free GOx and HRP cascade.** In order to characterize the GOx–HRP cascade reaction if both enzymes are free in solution, we measured the $ABTS^{+\bullet}$ concentration as function of time for different ratios of HRP to GOx in the presence of 1 mM glucose (Fig. 2). According to the measured kinetics of HRP, the maximum reaction velocities of 1, 2 and 20 nM HRP are 33, 65 and $650\,nM\,s^{-1}$, respectively, and are all greater than the reaction velocity of 1 nM GOx ($15.4\,nM\,s^{-1}$). There is a clear transient stage at the beginning in the case of 1 nM GOx with 1 nM HRP (Fig. 2a,b). Increasing the amount of HRP gradually shortens the transient time, leading to slightly higher concentrations of $ABTS^{+\bullet}$ at equal time points. The transient stage is almost undetectable when a 20-fold excess of HRP was used. The increase in reaction velocity of HRP causes a change in the product formation curve (Fig. 2b), which resembles the changes observed when GOx and HRP are placed on DNA scaffolds[9,10,17].

All of these reactions converge to a state in which the $H_2O_2$ consumption rate is equal to the reaction velocity of 1 nM GOx in 1 mM glucose ($15.4\,nM\,s^{-1}$). The fact that the final activity converges to an identical value demonstrates that the only

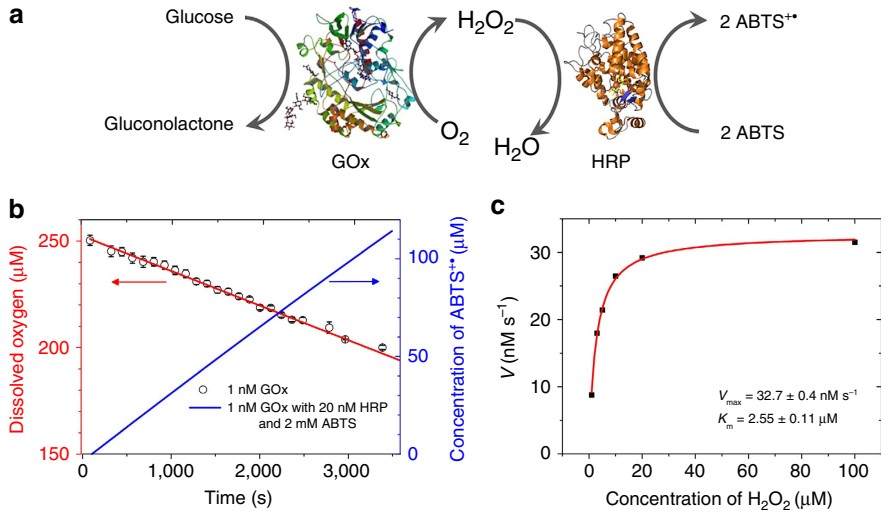

**Figure 1 | Individual kinetics of GOx and HRP.** (**a**) Schematic representation of GOx/HRP cascade. (**b**) Oxygen concentration in a sealed vessel containing 1 nM GOx and 1 mM glucose in PBS (red), and the concentration of $ABTS^{+\bullet}$ in PBS containing 1 nM GOx, 20 nM HRP, 1 mM glucose and 2 mM ABTS (blue) as function of time, error bars represent the s.d. of the three replicates. (**c**) HRP follows Michaelis–Menten kinetics; the activity of HRP was measured in PBS containing 1 nM HRP, 2 mM ABTS and different concentration of $H_2O_2$.

potential benefit of an improved transport is a shortening of the initial transient stage if the intrinsic kinetics of the involved enzymes do not change. This conclusion had been drawn in the literature decades ago when biochemists were first defining the concept of substrate channeling[22–25].

The time course of $ABTS^{+\bullet}$ production was modelled with a system of ordinary differential equations using the kinetic parameters measured individually for GOx and HRP (Supplementary Fig. 4). This model assumes that on the timescale of the experiment the produced $H_2O_2$ is well mixed. A detailed justification for this fact is given in our previous work[19]. For illustration, consider that, for a solution containing 1 nM HRP and 1 nM GOx, the average distance between the enzyme pairs is approximately 1 μm. Therefore, the $H_2O_2$ generated by GOx can diffuse from a GOx molecule to a HRP molecule in 0.6 ms (the diffusion coefficient of $H_2O_2$ is $1.71 \times 10^{-9}\,m^2\,s^{-1}$ at 25 °C (ref. 26)). As it takes about 10 s to mix the enzyme and substrate solution before the data acquisition commences, all simulation curves were shifted by 10 s. As shown in Fig. 2, the model fits the experimental results. The slight decrease in the final activity may be due to deactivation of GOx or product inhibition by the accumulating gluconolactone, but neither deactivation nor inhibition significantly influences the activity. Figure 2c shows that, in the cascade reaction, it takes 500 s to establish the steady state when 1 nM HRP is used. Increasing the concentration of HRP can shorten the transient time owing to a more rapid conversion of $H_2O_2$. As the maximum reaction velocity of the HRP ($V_{max}$ is 33 nM s$^{-1}$ for 1 nM HRP) always exceeds the production of $H_2O_2$ by GOx (15 nM s$^{-1}$ for 1 nM GOx), the $H_2O_2$ concentration converges to a constant value so that the actual reaction velocity of HRP equals 15 nM s$^{-1}$ (Supplementary Fig. 5).

We then carried out the reaction at lower concentrations of HRP, where the reaction rates are limited by the activity of HRP (Fig. 3). Reducing the HRP concentration from 0.4 to 0.2 nM now slows the rate of $ABTS^{+\bullet}$ production by half. The $H_2O_2$ concentration quickly exceeds the $K_m$ of HRP and continues to rise throughout the experiment (Supplementary Fig. 6), making the reaction rate of HRP approach its maximum value. The model also describes the process well despite the slight deactivation of HRP in the final stage.

The results above demonstrate that the diffusion of the intermediate substrate $H_2O_2$ is not the rate-determining step even in the initial stage of the free GOx–HRP cascade. Small molecules such as $H_2O_2$ with a diffusion coefficient on the order of $10^{-9}\,m^2\,s^{-1}$ diffuse so fast into the bulk solution that no channeling effect can be expected for such a cascade system. In addition, although the rate of product formation converges to a certain value, the concentration of the intermediate substrate may continue to increase if the downstream enzyme is rate limiting. Increasing the rate of the non-limiting step can only shorten the transient stage (which may appear to be 'enhanced throughput'), whereas improving the activity of the rate-limiting enzyme can significantly and permanently enhance the throughput.

**GOx–HRP conjugate.** In order to place GOx and HRP in close proximity to each other, we covalently conjugate GOx and HRP with a small molecular linker, sulfosuccinimidyl 4-(N-maleimidomethyl)cyclohexane-1-carboxylate (sulfo-SMCC), with a spacer length of 8.3 Å. As the free cysteine residues of the GOx are hard to access, we grafted more sulfhydryl groups from the lysine residues via N-succinimidyl 3-(2-pyridyldithio)propionate (SPDP) modification and subsequent dithiothreitol thiolation (Fig. 4a). HRP was purified by size exclusion chromatography prior to modification (Supplementary Figs 7 and 8). After the conjugation, the maximal distance between HRP and GOx is 2.3 nm if the flexible linker is fully stretched. The conjugate was separated by size exclusion chromatography on a HiLoad 16/600 Superdex 200 prep grade column (Supplementary Figs 9 and 10). The molar ratio of GOx and HRP in the conjugate is 0.94: 1, as determined from its absorbance at 280 and 403 nm (Supplementary Fig. 11). SDS–PAGE shows that the conjugate is a mixture of multimers with a different stoichiometry of GOx to HRP and that all HRP was conjugated with GOx (Fig. 4b). The activity of GOx was determined as 14.1 nM s$^{-1}$ for the amount of conjugate containing 1 nM HRP in 1 mM glucose. The $V_{max}$ and $K_m$ of the conjugated HRP were almost the same as those of free HRP, indicating that its activity as well as its affinity for $H_2O_2$ had not been affected by the conjugation (Fig. 4c). The specific activities

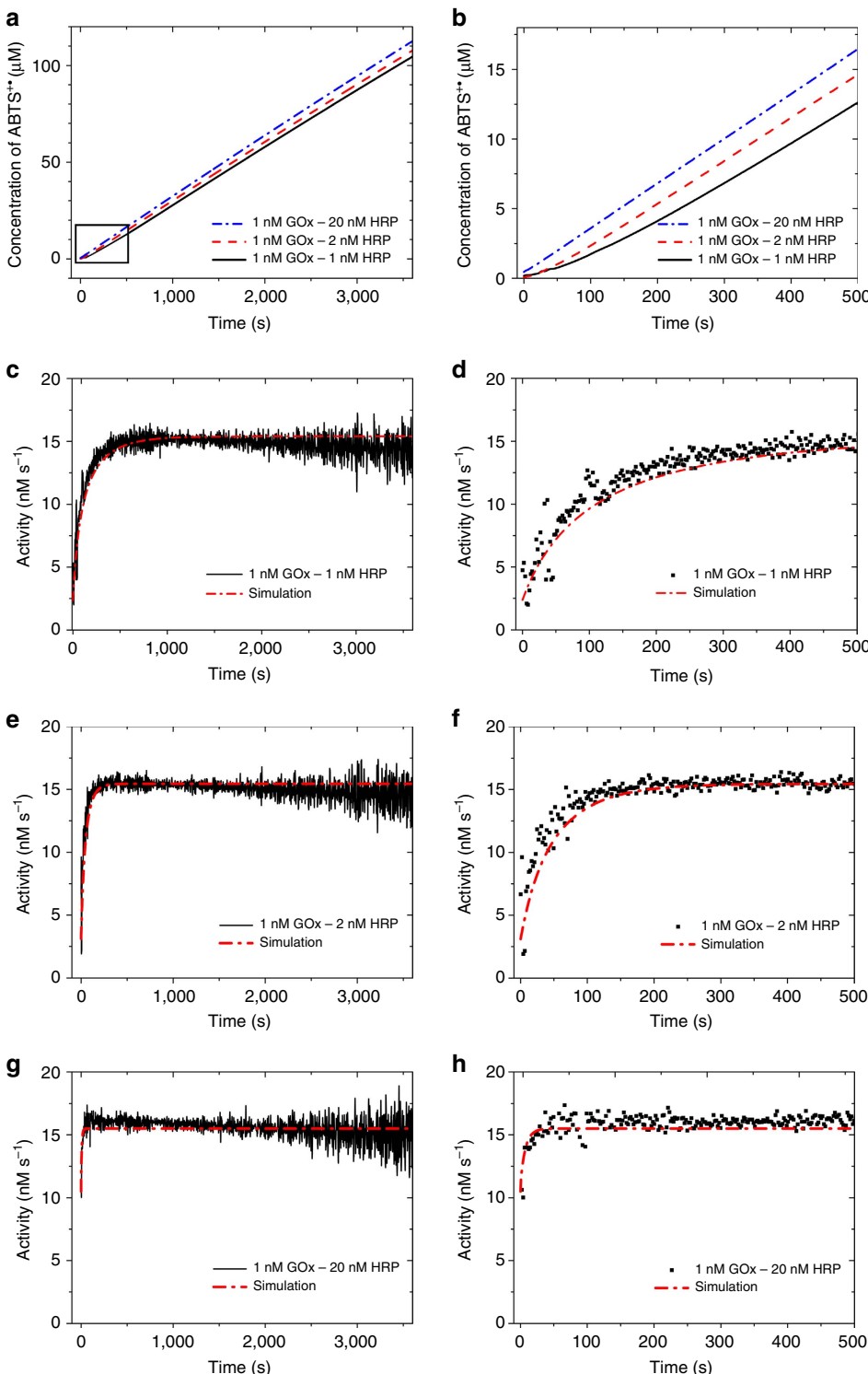

**Figure 2 | Free GOx and HRP cascade when GOx is the limiting enzyme.** (**a**) Concentration of ABTS$^{+\bullet}$ in the presence of 1 nM GOx and 1, 2 or 20 nM HRP and (**b**) the initial period of panel (**a**). (**c**,**e**,**g**), Activity of GOx–HRP cascade with respect to the consumption of H$_2$O$_2$ in the presence of 1, 2 and 20 nM HRP and (**d**,**f**,**h**) the initial periods of panels (**c**,**e**,**g**). All assays were independently replicated three times.

of GOx and HRP in the conjugate were not significantly different from their original value, respectively (Fig. 4d,e). We then carried out the cascade reaction with this conjugate (Fig. 4f). The catalytic process of the GOx–HRP conjugate can also be described by our model, without requiring additional terms for a substrate channeling effect. The steady state is reached after 500 s, and the final overall activity was limited by

the slower enzyme, GOx (Fig. 4g). It clearly demonstrates that simply linking two enzymes together or placing them close to each other has no benefits on the overall activity, that is, proximity is not the reason for the activity enhancement for the GOx–HRP cascade or other multienzyme system involving small-molecular intermediates with high diffusion coefficients.

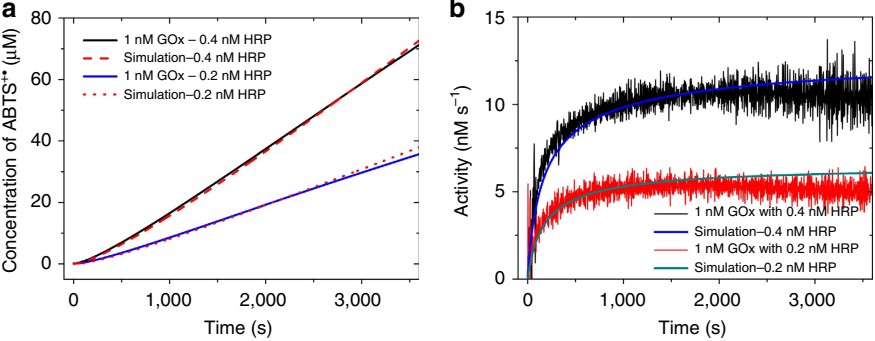

**Figure 3 | Free GOx and HRP cascade when HRP is the limiting enzyme.** (**a**) The time courses of ABTS$^{+\bullet}$ production and the simulation curves in the presence of 1 nM GOx and 0.2 nM or 0.4 nM HRP. (**b**) Activity of the GOx–HRP cascade with respect to the consumption of H$_2$O$_2$. The assays were independently replicated twice.

**Testing for substrate channeling with catalase (CAT) competition.** To further confirm that the proximity does not result in substrate channeling, we introduce varying concentrations of CAT to the GOx–HRP conjugate catalysed reaction (Fig. 5a). CAT consumes H$_2$O$_2$ in the bulk solution and diverts it away from HRP but should not affect the direct channeling of H$_2$O$_2$ from GOx to HRP. As shown in Fig. 5b, the addition of CAT can completely suppress the formation of ABTS$^{+\bullet}$, indicating that all the H$_2$O$_2$ enters the bulk solution and is accessible to the CAT. The time course of ABTS$^{+\bullet}$ can also be well described by our model augmented with a term describing the CAT-induced sequestering of H$_2$O$_2$ (Supplementary Figs 12–15). If a second term describing the direct channeling of a fraction of the GOx output is introduced, a fit to the data in Fig. 5b leads to the conclusion (with 95% confidence) that the directly channeled fraction is <0.5% (see Supplementary Information for details).

**pH dependence as a potential source of activity enhancement.** If the proximity effect is discarded, other explanations for the strongly enhanced throughput of the GOx–HRP cascade on DNA scaffolds are needed. We checked that the activity of GOx and HRP is not altered by the buffer used for DNA origami (Supplementary Fig. 16). Idan and Hess[18,27] suggested that the aggregation of multiple enzyme pairs on a large scaffold can lead to throughput enhancement. The colocalization of enzymes does ensure the fast consumption of intermediate and prevent it escaping[28], but the work presented by Fu et al.[11] clearly showed that enhancement occurs even if there are individual enzyme pairs on each scaffold and the enhancement effect is permanent. Lin and Wheeldon pointed out that attractive interactions between a DNA scaffold and the substrate can facilitate the transport of the substrate to the enzyme[29,30], and the model by Idan and Hess[18] suggested that the effect could be long-lasting. However, DNA does not attract the negatively charged substrate ABTS[31] and the throughput enhancement is caused by facilitated transport and should disappear as the concentration of the intermediate substrate builds up in the reaction vessel.

Recently, significant enhancement of throughput for the GOx–HRP cascade was reported in giant DNA structures. Linko et al.[32] encapsulated GOx and HRP in DNA origami tubes with dimensions of $33 \times 27 \times 60$ nm$^3$. The activity of GOx origami and HRP origami increased about threefold and fivefold, respectively. Zhao et al.[21] anchored different enzymes in DNA nanocages ($54 \times 27 \times 27$ nm$^3$) and found that many encapsulated enzymes showed much higher activities than their

free form. Specifically, the $k_{cat}$ for the HRP increased from 32 to 290 s$^{-1}$ after encapsulation, and other enzymes such as GOx, malate dehydrogenase and lactate dehydrogenase all showed increased activities. This suggests that the DNA itself was the major reason for the increase in the enzymatic activities. Zhao and colleagues suggested that the enhancement is because the hydration layer attracted by negatively charged DNA can stabilize the enzymes located around the highly ordered water molecules. However, the ordered water layer only contributes within a few angstroms[33,34], and the stabilization of the enzyme is not directly related to a higher activity.

We believe that the many negative charges on large DNA structures, and in particular on DNA origami, are very important for the activity of conjugated enzymes. Similar to polyelectrolyte films[35], the pH close to the surface of a DNA nanoplate is much lower than that in the bulk solution because the protons are attracted to the negatively charged interface, as illustrated in Fig. 6a. The relatively lower local pH will significantly shift the pH-activity profile of anchored enzymes, as demonstrated by Goldstein and colleagues 50 years ago. They immobilized trypsin on a water-insoluble polyanionic carrier and found that the pH-activity profile of immobilized trypsin shifted toward more alkaline pH values by 1–2.5 pH units, and this effect is highly depended on the ionic strength of the bulk solution[36]. A similar effect was also observed for a polyanionic derivative of chymotrypsin and papsin[37,38]. For the DNA origami, the surface pH is related to the bulk pH and the surface electrical potential $\psi$ by equation (1) (ref. 39):

$$pH_s = pH_b + \frac{F\Psi}{2.3RT} \qquad (1)$$

where pH$_s$ is the pH at the surface, pH$_b$ is the pH in the bulk solution, F is the Faraday constant, $R$ is the universal gas constant, and $T$ is the temperature.

Although the exact electric surface potential of DNA origami has not yet been reported, we can estimate its magnitude from similar situations. Knudsen et al.[40] measured the surface potential of a single-stranded DNA-grafted brush polymer by atomic force microscopy to be $-130$ mV. Wong and Melosh[41] estimated the electrostatic potential at a DNA-hybridized surface. For a DNA-grafting density of $3 \times 10^{13}$ cm$^{-2}$, the surface potential is also about $-130$ mV. The density of negative charges on DNA origami is significantly higher, although a theoretical description of the resulting surface potential is not straightforward[41]. We calculated the pH distribution near the surface of DNA origami in a pH 7.5 solution with 10 mM monovalent salt concentration by solving the Poisson

Boltzmann equation for an infinite sheet of uniform charge density of $36\,nm^{-2}$ (from ref. 21):

$$\nabla^2\Psi = -\frac{2FC_0}{\varepsilon}\sinh\left(-\frac{F\Psi}{RT}\right) \quad (2)$$

In equation (2), $C_0$ denotes bulk salt concentration, and $\varepsilon$ is the product of the vacuum permittivity constant and dielectric constant of water (see Supplementary Information for details). The pH profile is shown in Fig. 6b and suggests that, even at 2 nm from the scaffold surface, the pH is one unit lower than that of

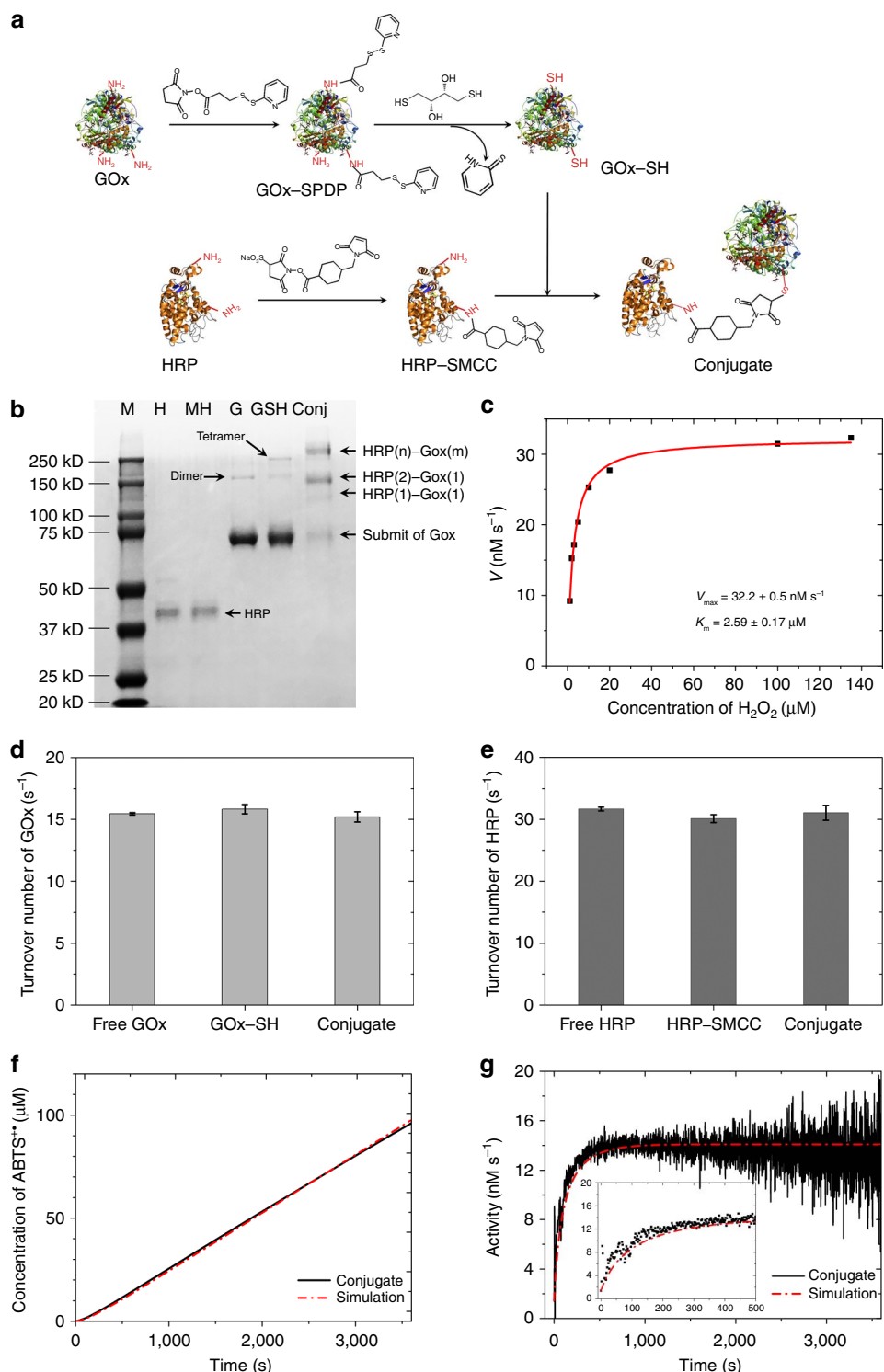

**Figure 4 | Synthesis and the kinetics of GOx–HRP conjugate. (a)** Synthesis of GOx–HRP conjugate. **(b)** SDS-PAGE analysis of the conjugate. The five lanes to the right of the standard marker correspond to native HRP, modified HRP (HRP-SMCC), native GOx, GOx–SH and the conjugate. Each lane is loaded with 1 μM protein. **(c)** Kinetics of HRP in the conjugate; the measurement was carried out with 1 nM conjugated HRP and 2 mM ABTS in PBS buffer. **(d)** Activities of free GOx, GOx-SH and the conjugated GOx. **(e)** Activities of free HRP, HRP-SMCC and the conjugated HRP. **(f)** The time courses of ABTS$^{+\bullet}$ production and the model prediction based on the known activity of conjugated GOx and the $V_{max}$ and $K_m$ of conjugated HRP. **(g)** The differential curve of panel **(f)** presenting the $H_2O_2$ consumption rate. Error bars in panels **(d)** and **(e)** represent the s.d. calculated from three replicates.

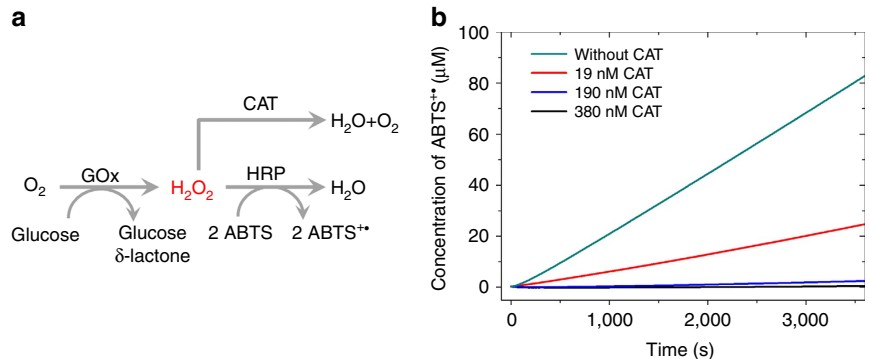

**Figure 5 | GOx–HRP cascade with catalase competition.** (**a**) Schematic of the GOx–HRP cascade coupled with a competing reaction catalysed by catalase. (**b**) The suppression of ABTS$^{+\bullet}$ production from the GOx–HRP conjugate by addition of catalase; here the reaction velocity of GOx is 11.8 nM s$^{-1}$ and $V_{max}$ of HRP is 25.3 nM s$^{-1}$. For the free GOx–HRP experiment, see Supplementary Fig. 15.

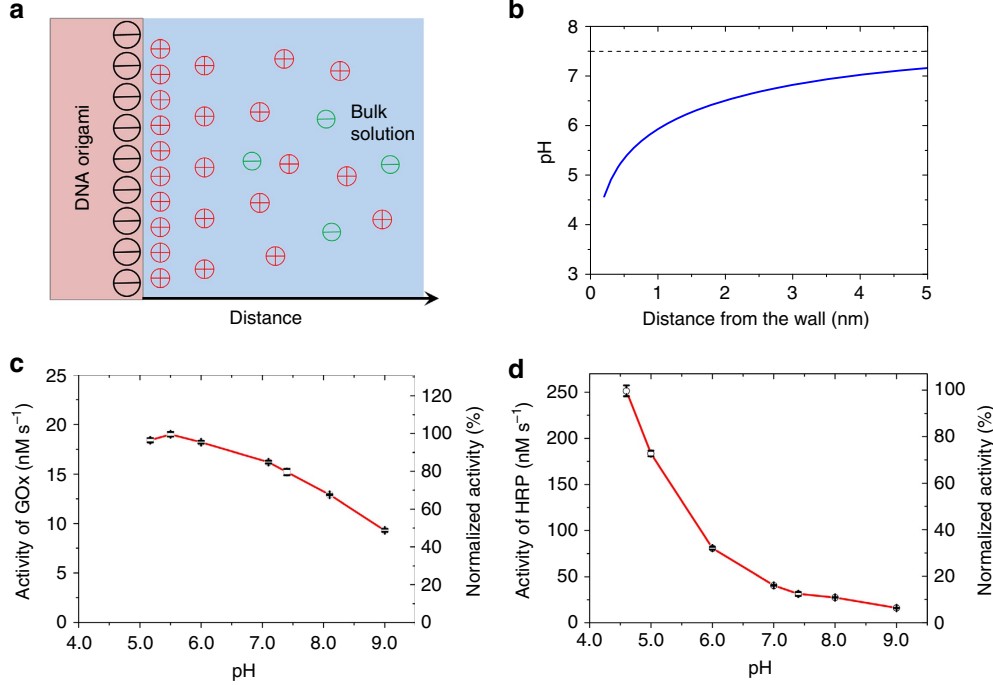

**Figure 6 | Influence of DNA on the enzymatic activity.** (**a**) Schematic of the distribution of positive ions near the surface of DNA origami. (**b**) pH-distance profile near the DNA origami. (**c**) pH-activity profile of GOx in 1 mM glucose and (**d**) pH-activity profile of HRP in 200 μM H$_2$O$_2$ and 2 mM ABTS. Error bars represent the s.d. of three replicates.

the bulk solution. Steric effects and penetration of counterions into the DNA layer will have large effects at these high charge densities. As a result, the pH profile in Fig. 6b is only a very rough approximation. Its only message is that any enzyme bound to the charged surface spends a non-negligible amount of time in a lower pH regime; this may in turn affect its catalytic activity.

To test whether GOx and HRP activity may be affected, we determined the pH dependence of HRP and GOx activity. Both HRP and GOx show an increased maximal turnover rate in more acidic conditions (Fig. 6c,d and Supplementary Fig. 17). The activity of 1 nM of HRP dramatically increases from 32 to 250 nM s$^{-1}$ when the pH changes from 7.4 to 4.6, which is very close to the observation by Zhao *et al.*[21] for HRP activity enhancement in a DNA nanocage. Also, according to the studies by Goldstein *et al.*[36–38] if the GOx and HRP are anchored on a polyanionic carrier that can decrease the local pH by 2 units (from 7.5 to 5.5), then their activity will increase by 1.2- and 4-fold, respectively, leading to an enhanced overall

activity. In the previous publications[10,11,17], if the HRP is the limiting enzyme, the lower local pH will significantly enhance the cascade activity; if the GOx is the limiting enzyme, the combination of the increased activity of GOx and the shortened transient time by the much more active HRP will also result in enhanced cascade activity.

The precise localization of enzyme cascades on DNA scaffolds and the resulting enhancement of the cascade throughput has been one of the most exciting advances in nanobiotechnology in recent years. However, the origin of the observed throughput enhancement, and the contribution of proximity to it, has been debated. Using the canonical GOx–HRP cascade, we demonstrated that the final activity of the cascade reaction is determined by the slower enzyme, which means that throughput enhancement beyond the initial phase requires an increase in the activity of the rate-limiting enzyme. By conjugating GOx and HRP, measuring the kinetics of the individual enzymes and the conjugate and measuring the kinetics of the conjugate in the

presence of an enzyme competing for the intermediate substrate, we were able to experimentally demonstrate that proximity has no effect on the final cascade activity, thereby confirming theoretical predictions. We further show that the activity of GOx and HRP increases with decreasing pH and suggest that the origin of the throughput enhancement of the GOx–HRP cascade on a DNA scaffold lies in the highly negatively charged DNA lowering the local pH experienced by the enzymes. Based on these insights, three questions should be carefully evaluated in order to support the claim that proximity causes activity enhancement in cascades: is there a steady state and has it been reached? which enzyme is the rate-limiting one? did the individual activity of this enzyme change? Our findings provide a new insight into DNA scaffold multienzyme systems and point out a novel strategy to optimize the pH in the microenvironment of enzymes.

## Methods

**Materials.** GOx from *Aspergillus niger* (type VII), HRP, D-glucose, $H_2O_2$ (30 wt.% in water) and ABTS were purchased from Sigma Aldrich Co. LLC, St Louis, MO, USA; SPDP and sulfo-SMCC were purchased from ThermoFisher Scientific, Inc., USA. For all the experiments, the purchased HRP was first purified by size exclusion chromatography to remove the impurities (Supplementary Figs 7 and 8). The RZ value ($A_{403}/A_{280}$) of purified HRP is $2.5 \pm 0.1$.

**Enzymatic assays.** All of the enzyme quantifications were carried out on a ultraviolet–visible spectrophotometer (Evolution 201, Thermo Scientific, USA). The extinction coefficients for GOx at 280 and 450 nm are $2.67 \times 10^5$ and $2.61 \times 10^4 \, M^{-1} \, cm^{-1}$, respectively. The extinction coefficient for HRP at 403 nm is $1.0 \times 10^5 \, M^{-1} \, cm^{-1}$. Enzymatic activities were measured from the changes in absorbance at 415 nm (or 600 nm) by the spectrophotometer (for the final product ABTS radical cation, $\varepsilon_{415 \, nm} = 3.6 \times 10^4 \, M^{-1} \, cm^{-1}$ and $\varepsilon_{600 \, nm} = 1.1 \times 10^4 \, M^{-1} \, cm^{-1}$).

For the GOx kinetic assay, 20 µl of 50 nM GOx was added to 980 µl of substrate solution (in PBS buffer, pH 7.4) to initiate the reaction. The final assay solution contains 2 mM ABTS, 1 mM glucose, 1 nM GOx and 20 nM HRP. The increase in absorbance at 415 nm was recorded. The activity was computed from the slope of the absorbance versus time curve during the first 1 min.

The consumption of oxygen by GOx was measured by a dissolved oxygen meter (Mettler Toledo, FiveGo) in PBS buffer containing 1 mM D-glucose in a sealed vessel. We placed the probe of the dissolved oxygen meter in a glass vessel fully filled with 3 ml substrate buffer. As soon as 10 µl of 300 nM GOx was added to the solution (final concentration is 1 nM), the vessel is immediately sealed with a piece of plastic paraffin film. The oxygen concentration was recorded for 1 h under magnetic stirring.

For the HRP activity assay, 20 µl of 50 nM HRP, 20 µl of 10 mM $H_2O_2$ and 20 µl of 2 mM ABTS solution was added to 940 µl of PBS buffer (10 mM, pH 7.4). The absorbance at 415 nm (or 600 nm) was recorded for the first 2 min.

For the GOx–HRP cascade reaction, the final assay solution contains 1 nM GOx, 1 nM HRP, 2 mM ABTS and 1 mM D-glucose. The absorbance at 600 nm was recorded for 1 h.

All the assays were replicated at least twice. We also carried out the experiments with different batches of samples. For the time-course curve, as the zero point drift and the time offset for mixing slightly vary from parallel experiments, we do not average them but analyse them individually.

All the assays were carried out at $23 \pm 1 \, °C$.

**Synthesis of HRP and GOx conjugate.** The HRP and GOx was conjugated with a bi-functional linker, sulfo-SMCC. As the two free cysteine residues on GOx are hard to access, free sulfhydryl groups were first grafted from the lysine residues on the surface of GOx. In a typical reaction, 40 µl of 80 mM SPDP solution (in dimethyl sulfoxide) was added into 1 ml of 40 µM GOx (in 50 mM phosphate buffer, pH = 7.5) with a ratio of 80:1. After stirring for 1 h at room temperature, 20 µl of 1 M dithiothreitol was added to the mixture for 30 min. The excess linker and dithiothreitol were removed by passing the solution twice through a Hitrap desalting column (GE Healthcare Bio-Sciences, PA, USA). The concentration of GOx–SH was determined from its absorbance at 280 nm.

HRP was functionalized via the maleimide group with sulfo-SMCC. Typically, 80 µl of 40 mM sulfo-SMCC solution (in 50 °C water) was added dropwise into 500 µl of 40 µM HRP solution (50 mM phosphate buffer, pH 7.5). The mixture was stirred for 1 h at room temperature. The HRP–SMCC was purified by passing it through a Hitrap desalting column twice. The concentration of HRP–SPDP was determined by its absorbance at 403 nm.

For the conjugation of GOx and HRP, GOx–SH solution was added into the purified HRP–SMCC solution at a molar ratio of 1: 1. The mixture was incubated at room temperature for 1 h and stored at 4 °C before separation. The conjugated enzymes were separated by size exclusion chromatography with a HiLoad 16/600 Superdex 200 prep grade column on an ÄKTA system.

**Numerical simulation of GOx–HRP coupled reaction.** As the $H_2O_2$ (diffusion coefficient: $1.71 \times 10^{-9} \, m^2 \, s^{-1}$) can diffuse the average distance between enzyme pairs (about 1 µm for the system containing 1 nM GOx and 1 nM HRP) in 0.6 ms, the $H_2O_2$ solution can be considered well mixed on the timescale of the experiment, and the cascade reaction is determined by the kinetics of GOx and HRP. The production rate of $H_2O_2$ depends on the GOx activity (equation (3)), the consumption of $H_2O_2$ is described by Michaelis–Menten kinetics of HRP (equation (4)), the concentration of $H_2O_2$ in the solution is described by equation (5) and the production of ABTS$^{+\bullet}$ follows equation (6). The initial $H_2O_2$ concentration is zero (equation (7)).

$$r_1 = k_{cat,1}[E_1] \tag{3}$$

$$r_2 = \frac{V_{max,2}[I]_t}{[I]_t + K_{m,2}} \tag{4}$$

$$[I]_t = r_1 t - \int_0^t r_2 \, dt \tag{5}$$

$$[P] = 2 \cdot \int_0^t r_2 \, dt \tag{6}$$

$$[I]_{t_0} = 0 \tag{7}$$

where $[I]_t$ is the concentration of the intermediate ($H_2O_2$) at time $t$; $r_1$ is the generation rate of $H_2O_2$ by GOx, which is approximately constant in our experiment; $k_{cat,1}$ is the turnover number of GOx; and $V_{max,2}$ and $K_{m,2}$ are the maximal velocity and Michaelis constant of HRP, respectively.

**Data availability.** Data supporting the findings of this study are available within the article and its Supplementary Information Files and from the corresponding author upon reasonable request.

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

## Acknowledgements

We thank Miss Kristen Garcia for the purification of the conjugate. This work was supported by the Defense Threat Reduction Agency under award number HDTRA 1-14-1-0051.

## Author contributions

Y.Z. and H.H. conceived and designed the research. Y.Z. performed all the experiments. S.T. performed the statistical analysis and analysed the pH profile at the surface of DNA origami. All the authors discussed the results and commented on the manuscript.

## Additional information

**Competing financial interests:** The authors declare no competing financial interests.

