## [Peer Review File · Nature Communications]

Reviewers' comments:

Reviewer #1 (Remarks to the Author):

This work by Zhang, Tsitkov, and Hess is timely and important, demonstrating that the commonly held position that proximity between two enzymes in a coupled reaction can enhance steady state cascade reaction rate is not true. Or at least is not true in the context of the model step-two cascade of glucose oxidase (GOx) and horseradish peroxidase (HRP) with the current technologies for spatial organization (e.g., protein and DNA molecular scaffolds and site selective protein crosslinking). The manuscript is well written and clear, the conclusions are justified, and the presented results support the manuscript's title.

The idea that substrate channeling by proximity has significant effects on cascade reaction rates by enabling faster diffusion of cascade intermediates from one enzyme to the next is gaining traction in the current literature. A debate of the effects of proximity was prevalent in the literature more than a decade ago when biochemists were first defining the concept of channeling (see refs [1-4] among others). The general conclusion (supported by both experiments and theoretical modeling) was that the overall rate of a cascade reaction is not effected by substrate channeling. This is an important concept that needs to be addressed again in the context of new DNA nanotechnologies that enable the nm-scale positioning of cascade enzymes.

Using the model GOx-HRP cascade Zhang, Tsitkov, and Hess use standard protein crosslinking to create a two-enzyme system where the upstream and downstream active sites are <3 nm from one another (Fig. 4). Kinetic analysis of the individual enzymes shows that their activities are not disrupted by the conjugation (Fig. 1 and 4.) and analysis of the cascade (both experimentally and with kinetic modeling) shows that the time course of the coupled enzyme reaction is identical with and without enzyme organization (Fig. 4). A standard test for channeling by a competing side reaction supports this conclusion (Fig. 5).

The manuscript provides two possible explanations for the previously reported enhancements in cascade rate observed with GOx and HRP in spatially organized DNA nanostructures: 1) that previous measurements were made during the transient time of the cascade when reaction velocities have not yet reached steady state; and 2) that DNA nanostructures create a low pH environment in close proximity to the enzyme and alter the apparent enzyme kinetics. The first explanation is supported by the author's previous modeling efforts,[5] as well as the transient time analysis of coupled enzyme reactions.[3, 6] The second explanation is new and has not been considered by current works in the field.[7-9](among others) The second point is not explicitly addressed in the current manuscript as control experiments with GOx/HRP-DNA nanostructures are not tested. A direct comparison of the GOx-HRP conjugated system presented here and one or more of the DNA nanostructures that assemble a GOx-HRP cascade would significantly add to the work. Such experiments are somewhat outside of the scope of the work, which stands on its own merits. In the absence of a direct comparison, the subtitle starting on line 208 should be more explicit that DNA nanostructures are a potential source of catalytic enhancement.

Overall the manuscript is clear, but the following comments should be addressed:

- 1) The measured k_{cat} values of GOx and HRP are lower than reported values in the cited references. For example, see ref 11 of the manuscript that reports a GOx k_{cat} of ~ 300 1/s and ref. 25 that reports a HRP k_{cat} of > 700 1/s for ABTS. Please discuss/explain.
- 2) The author's previously published models (refs 18 and 19 of the paper) suggest that the duration of transient increase in cascade rate is a "normally short-lived transient effect (on the millisecond timescale)", but in the current work the transient effect lasts for more than 200 seconds (see Fig. 2). Please clarify/explain.
- 3) What could account for the difference in oxygen consumption and ABTS formation if the stoichiometry is 1:2? (see line 87).
- 4) For clarity, please expand on the comment beginning on line 101 that "The reaction velocity of

HRP is greater than the reaction velocity of 1 nM GOx...”

- 5) More than one reference should be provided for the sentence ending on line 108.
- 6) The conclusion stated in the paragraph beginning on line 109 has previously been made and should be discussed and cited. See refs. [1-4] among others.
- 7) Including specific values of HRP and GOx activities in the sentence beginning on line 133 would be helpful.
- 8) Why are the activities of GOx and HRP reported in the legend of Fig 5 lower than the values reported in Fig 4?
- 9) Ref 13 of the manuscript is a review paper, only experimental examples should be cited on line 274. Also, it is not clear why HRP is rate limiting in ref. 11. The ratio of GOx:HRP is 1:1, similar to the experiments shown here in Figs 2 and 4 where HRP is not rate limiting. Please clarify/explain.
- 10) Ref. 21 of the manuscript is discussed at length with respect to increases in k_{cat} in DNA nanostructure environments. Are the other, non-HRP, enzymes studied in ref. 21 also pH dependent?
- 11) Some discussion should be added to explain how the results of Fig 6 can account for the change in GOx-HRP cascade enhancement with changes in interenzyme distance described in ref. 11 of the manuscript. If the DNA nanostructure alters the local pH, thus increasing the apparent kinetics of HRP then why is there an observed distance-dependent effect?
- 12) The number of technical and biological replicates should be clearly stated in each figure legend.

Nice paper.

References.

- [1] Spivey, H. O., Ovadi, J., Substrate channeling. *Methods* 1999, 19, 306-321.
- [2] Ovadi, J., Tompa, P., Vertessy, B., Orosz, F., et al., Transient-time analysis of substrate-channelling in interacting enzyme systems. *Biochem J* 1989, 257, 187-190.
- [3] Elcock, A. H., Huber, G. A., McCammon, J. A., Electrostatic channeling of substrates between enzyme active sites: comparison of simulation and experiment. *Biochemistry-Us* 1997, 36, 16049-16058.
- [4] Elcock, A. H., McCammon, J. A., Evidence for electrostatic channeling in a fusion protein of malate dehydrogenase and citrate synthase. *Biochemistry-Us* 1996, 35, 12652-12658.
- [5] Idan, O., Hess, H., Origins of Activity Enhancement in Enzyme Cascades on Scaffolds. *ACS Nano* 2013.
- [6] Trujillo, M., Donald, R. G. K., Roos, D. S., Greene, P. J., Santi, D. V., Heterologous expression and characterization of the bifunctional dihydrofolate reductase-thymidylate synthase enzyme of *Toxoplasma gondii*. *Biochemistry-Us* 1996, 35, 6366-6374.
- [7] Fu, J. L., Liu, M. H., Liu, Y., Woodbury, N. W., Yan, H., Interenzyme Substrate Diffusion for an Enzyme Cascade Organized on Spatially Addressable DNA Nanostructures. *J Am Chem Soc* 2012, 134, 5516-5519.
- [8] Fu, J. L., Yang, Y. R., Johnson-Buck, A., Liu, M. H., et al., Multi-enzyme complexes on DNA scaffolds capable of substrate channelling with an artificial swinging arm. *Nature Nanotechnology* 2014, 9, 531-536.
- [9] Wilner, O. I., Weizmann, Y., Gill, R., Lioubashevski, O., et al., Enzyme cascades activated on topologically programmed DNA scaffolds. *Nature Nanotechnology* 2009, 4, 249-254.

Reviewer #2 (Remarks to the Author):

The authors report kinetic measurements and simulation of the GOX-HRP cascade reaction, which is the most prominent example of synthetic distance-dependent enzyme networks studied so far. Owing to the high relevance of the GOX-HRP system for motivating research activities in nanobiotechnology (which is not limited to DNA-arranged enzymes), the key finding that

proximity does not contribute to activity enhancement would, in principle, qualify the paper for publication in Nature Communication. However, the manuscript appears to be not clearly focussed and has also technical flaws.

The lack of clear focus is evident from the introductory part (p3, line 47 ff) which aims to justify the relevance of the present work. The authors argue that „differences in the reported activity enhancements (from 1.5 to larger 15-fold)“ would justify their work. In fact, all cited examples used different assembly strategies, and thus, it is not too surprising that differences in activity enhancements occur. While this referee strongly supports the notion that postulated mechanisms should lead to quantitatively comparable results, it is only fair to note that different assembly systems would of course lead to different extents of an effect.

Differences in the reported activity enhancements might also stem from insufficient characterization or purification of the derivatized enzymes and complexes; enzyme modification often leads to changed activities and remaining free enzymes can distort activity measurements (for a discussion, see Timm, Angewandte 2015, 6745) - While this reviewer agrees that necessary controls were not always conducted in previous reports on enzyme cascades, it seems important to note that some of the „1.5 to larger 15-fold“ differences could easily be explained by technical imprecisions.

The authors argue that several synthetic multi-enzymes did not at all reveal enhanced activity, citing refs 16, 17. This is a bit misleading because ref 17, for instance, concerned entirely different enzymes (in addition to a completely different assembly method). It is thus unclear whether the authors want to address just the HRP-GOX cascade or whether the paper is meant as a general assessment of proximity effects in multienzymes? In the latter case, other papers which reported proximity-enhanced activities of DNA-linked enzyme cascades (e.g., Niemeyer, ChemBiochem 2002, 242 or Timm, Angewandte 2015, 6745) should also be cited.

Furthermore, the authors conclude a contradiction from their literature review that „the reported activity enhancements appear to exist indefinitely“ by citing refs .11,15,21. Having checked the three papers, this referee does not understand this argumentation.

The data shown in Figures 1 - 3 appear to be sound. Michaelis-Menten parameters of HRP have certainly been measured in several other studies before - could the authors please compare their data with previously reported ones for HRP and GOX.

The experimental details for determination of the Michaelis-Menten parameters of GOX are too scarce; please describe precisely which instrumentation was used.

Can the authors exclude that the kinetic data are inaccurate due to the huge amounts of impurities (>80 % !) evident from Fig. S7 ? What is the nature of these impurities? Were the data shown in Figures 1-3 generated with purified or non-purified HRP?

Fig. 5: Which cascade was studied here? Isolated enzymes or the chemical conjugates? I would expect that both systems should be compared.

Fig. S10: How were the spectra normalized? This kind of representation is highly prone to artefacts when different amounts/concentrations of proteins are used. Hence, protein needs to be thoroughly quantified before - how has this been done here? A control should be included in which equal amounts of pure GOX and HRP are mixed with each other.

Fig. 6 and the related discussion: The fact that origami samples are typically conducted in TEMg buffer (containing 12.5 mM MgCl₂) seems to be neglected. Divalent Mg cations should significantly alter the surface charge of polyelectrolytes and also affect the changes in pH. The authors used „10 mM salt“ (sodium chloride?) but magnesium ions have significantly different solvation

properties. In addition to taking this into account for their simulations, the authors should also experimentally verify that the activity of their enzymes is not affected by the TEMg buffer (20 mM Tris, 2 mM EDTA, 12.5 mM MgCl₂, pH = 7.6)

Fig. 6c: The activity of GOX appears to be much higher than that shown in Fig. 4d. How are these differences explained?

Based on the author's conclusions, I am still puzzled and, for instance, do not understand how results of Ref 11 can be explained where DNA-coupled GOX and HRP enzymes were compared to each other. Furthermore, the enhancement of enzyme activity through conjugation with DNA was already well documented from previous studies (see Timm, mentioned above; Fruk, *Chem. Eur. J.* 2006, 7448; Glettenberg, *Bioconj. Chem.* 2009, 969; Rudiuk, *Angewandte* 2012, 12694). Hence, the novelty of findings might be disputable and the question remains what we can learn from this paper for future studies? Do the authors expect that proximity does not contribute to activity enhancement of other cascades as well?

Reviewer #3 (Remarks to the Author):

Comments for Manuscript:

Proximity Does Not Contribute to Activity Enhancement in the Glucose Oxidase-Horseradish Peroxidase Cascade

Yifei Zhang, Stanislav Tsitkov, Henry Hess

This manuscript presents a detailed look into explaining the mechanism of activity improvement of the glucose oxidase (GOx) and horseradish peroxidase (HRP) enzyme cascade on DNA scaffolds. Importantly, this work provides some direct investigation into the cause of improved activity due to enzyme localization to a DNA scaffold. Up until now, these improvements have been attributed to a substrate channelling or proximity effect due to the co-localization of the two enzymes onto the DNA scaffold. However, through a series of computational and enzymatic experiments examining the effects of varying the limiting enzyme, adding a competitive enzyme and fusing the enzymes via a different linker, the authors show that the increased throughput observed in previously published reports on the GOx-HRP cascade on DNA scaffold is not due to a proximity effect, but rather is a result of the local environment of low pH created by the negatively charged DNA backbone.

Page 2, Line 21: The authors appear to generalize the rate enhancements of enzymes on DNA scaffolds as a result of the local lower pH, but do not appear to provide sufficient evidence that this applies to enzymes other than the glucose oxidase and horseradish peroxidase cascade. While GOx and HRP may be the most common enzymes to use in demonstrating a cascade on a DNA scaffold, it is not the only enzyme set used. It may be more appropriate to conclude that the presented results 'overturn' the general proximity effect for the GOx HRP pair on DNA scaffolds, and also 'challenges' the accuracy of this in other cascades. This effect may indeed apply to other enzymatic cascades tethered to DNA, however without explicit examination this cannot yet be stated. A similar comment can be applied to the usage on page 3, line 69.

Page 3, Line 51: Please clarify how a transient effect can be extended if it only has an influence until the intermediate reaches a significant concentration. Also please clarify how aggregated enzymes, attractive interactions between intermediate and scaffold, as well as competing reactions can all provide the same effect, when the latter is later shown to be deleterious to reaction throughput.

Page 11: Catalase, a competing reaction to GOx and HRP, is described as suppressing formation of ABTS+• in both the conjugated and free enzyme forms, but on page 12, line 212 the authors reference papers by Idan and Hess that have suggested that competing reactions can lead to throughput enhancements. Please clarify how both effects are possible.

Page 13, Figure 6c: If only the activity of HRP increases at lower pH (free glucose oxidase sharply

loses activity below pH 5.5), how does that fit with the proposed effect of the reduced pH at the DNA backbone being responsible for activity improvements? Also, to be more comparable to the other data presented here and to that presented in the referenced articles, the pH-activity profile for GOx may be more informative if tested at 1 mM glucose. This will demonstrate the effect of pH on the conditions used with the DNA scaffold, and also be more representative to the proposed effect.

An experiment I would like to see included in the manuscript is a pH activity (or cascade throughput rate) profile of the GOx-HRP conjugate. This could likely most directly show the pH dependence of the cascade and also provide the appropriate enzyme proximity and stoichiometry compared to the cascade on a DNA scaffold.

Minor notes:

Page 1, Line 19: '...production by the GOx-HRP conjugate,...'

Page 3, Line 68: '...where the enzyme is located.'

Page 5, Line 100: '...concentration as a function of time for different ratios of HRP to GOx...'

Page 15, Line 297: 'sulphonic'

This work provides a nice, detailed study and presents some mechanistic insight to the cause of activity improvements in cascades tethered to DNA scaffolds. With the minor additional experiments and text clarifications, I would recommend this manuscript for publication.

Responses to The Reviewer's Comments

Reviewers' comments:

Reviewer #1 (Remarks to the Author):

This work by Zhang, Tsitkov, and Hess is timely and important, demonstrating that the commonly held position that proximity between two enzymes in a coupled reaction can enhance steady state cascade reaction rate is not true. Or at least is not true in the context of the model step-two cascade of glucose oxidase (GOx) and horseradish peroxidase (HRP) with the current technologies for spatial organization (e.g., protein and DNA molecular scaffolds and site selective protein crosslinking). The manuscript is well written and clear, the conclusions are justified, and the presented results support the manuscript's title.

The idea that substrate channeling by proximity has significant effects on cascade reaction rates by enabling faster diffusion of cascade intermediates from one enzyme to the next is gaining traction in the current literature. A debate of the effects of proximity was prevalent in the literature more than a decade ago when biochemists were first defining the concept of channeling (see refs [1-4] among others). The general conclusion (supported by both experiments and theoretical modeling) was that the overall rate of a cascade reaction is not effected by substrate channeling. This is an important concept that needs to be addressed again in the context of new DNA nanotechnologies that enable the nm-scale positioning of cascade enzymes.

Using the model GOx-HRP cascade Zhang, Tsitkov, and Hess use standard protein crosslinking to create a two-enzyme system where the upstream and downstream active sites are <3 nm from one another (Fig. 4). Kinetic analysis of the individual enzymes shows that their activities are not disrupted by the conjugation (Fig. 1 and 4.) and analysis of the cascade (both experimentally and with kinetic modeling) shows that the time course of the coupled enzyme reaction is identical with and without enzyme organization (Fig. 4). A standard test for channeling by a competing side reaction supports this conclusion (Fig. 5).

The manuscript provides two possible explanations for the previously reported enhancements in cascade rate observed with GOx and HRP in spatially organized DNA nanostructures: 1) that previous measurements were made during the transient time of the cascade when reaction velocities have not yet reached steady state; and 2) that DNA nanostructures create a low pH environment in close proximity to the enzyme and alter the apparent enzyme kinetics. The first explanation is supported by the author's previous modeling efforts,[5] as well as the transient time analysis of coupled enzyme reactions.[3, 6] The second explanation is new and has not been considered by current works in the field.[7-9](among others) The second point is not explicitly addressed in the current manuscript as control experiments with GOx/HRP-DNA nanostructures are not tested. A direct comparison of the GOx-HRP conjugated system presented here and one or more of the DNA nanostructures that assemble a GOx-HRP cascade would significantly add to the work. Such experiments are somewhat outside of the scope of the work, which stands on its own

merits. In the absence of a direct comparison, the subtitle starting on line 208 should be more explicit that DNA nanostructures are a potential source of catalytic enhancement.

Overall the manuscript is clear, but the following comments should be addressed:

1) The measured k_{cat} values of GOx and HRP are lower than reported values in the cited references. For example, see ref 11 of the manuscript that reports a GOx k_{cat} of ~ 300 1/s and ref. 25 that reports a HRP k_{cat} of > 700 1/s for ABTS. Please discuss/explain.

Response: First of all, we would like to appreciate the reviewer for this in-depth and highly specialized comments. We agree that the k_{cat} values of GOx and HRP always vary from paper to paper. However, it's worth noting that the absolute activity does not affect our discussion and conclusion. We carefully compared the activity of free enzymes and conjugated enzymes, and all data are consistent and fit well with the simulation results.

For GOx, we measured its initial activity in various concentrations of glucose and found that the k_{cat} was 250 s^{-1} (close to the number provided by the reviewer) and the K_m was 15 mM (see revised Supplementary Information, **Supplementary Figure 1**). The activity we presented in the manuscript was determined in 1 mM glucose, and therefore the apparent activity is 15 nM/s. For the k_{cat} of HRP, at least three factors may affect the activity of HRP. The most important is the source and type of HRP (HRP has many isomers with different catalytic capacity). Secondly, HRP is a bi-substrate enzyme which follows the Ping-Pong mechanism, the apparent k_{cat} and K_m is also influenced by the concentration of ABTS. The last factor is the shelf-time and the purity of ABTS since it is sensitive to oxygen and light. The combination of these factors will cause the difference between this study and ref 25. Notably, the k_{cat} of $32.7 \pm 0.4 \text{ s}^{-1}$ and K_m of $2.55 \pm 0.11 \text{ }\mu\text{M}$ in our paper are the same as those in ref 21 (they obtained a k_{cat} of 32 s^{-1} and K_m of $2.3 \text{ }\mu\text{M}$).

2) The author's previously published models (refs 18 and 19 of the paper) suggest that the duration of transient increase in cascade rate is a "normally short-lived transient effect (on the millisecond timescale)", but in the current work the transient effect lasts for more than 200 seconds (see Fig. 2). Please clarify/explain.

Response: These two publications aim to explain the potential benefits from the proximity effect of cascade enzymes. However, the ACS Nano paper (ACS Nano, 2013, 7, 8658–8665) calculated the "channeling time", which means the characteristic time when the fraction of intermediate supplied to the second enzyme from the bulk solution becomes equal to that from direct transport. In our manuscript here, the transient effect refers to the time for the reaction rate of the second enzyme reaches that of the first enzyme (if $V_{max,2} > V_1$). To avoid confusion, we have changed "short-lived transient effect" to "short-lived effect" on **Page 3, line 57**.

3) What could account for the difference in oxygen consumption and ABTS formation if the stoichiometry is 1:2? (see line 87).

Response: The slight difference in oxygen consumption ($54 \text{ }\mu\text{M}$) and ABTS^+ formation ($112 \text{ }\mu\text{M}$) results mainly from the limited accuracy of oxygen meter.

4) For clarity, please expand on the comment beginning on line 101 that "The reaction velocity of HRP is greater than the reaction velocity of 1 nM GOx..."

Response: We have expanded this sentence as below (**Page 5, line 110-113**):

"According to the measured kinetics of HRP, the maximum reaction velocities of HRP are 33 nM s⁻¹, 65 nM s⁻¹ and 650 nM s⁻¹ for 1 nM, 2 nM and 20 nM HRP, respectively, and are all greater than the reaction velocity of 1 nM GOx (15 nM s⁻¹)."

5) *More than one reference should be provided for the sentence ending on line 108.*

Response: Thanks, we have added more references here.

Page 5, line 118: "... which resembles the changes observed when GOx and HRP are placed on DNA scaffolds.^{9, 10, 17}"

6) *The conclusion stated in the paragraph beginning on line 109 has previously been made and should be discussed and cited. See refs. [1-4] among others.*

Response: We thank the reviewer for providing these references. We have cited them in the revision.

Page 7, line 130: "This conclusion had been drawn in the literature decades ago when biochemists were first defining the concept of substrate channeling.²⁴⁻²⁷"

7) *Including specific values of HRP and GOx activities in the sentence beginning on line 133 would be helpful.*

Response: Thanks. We have added the specific activity of HRP and GOx in the revision:

Page 7, line 146: "Since the maximal reaction velocity of the HRP (V_{\max} is 33 nM s⁻¹ for 1 nM HRP) always exceeds the production of H₂O₂ by GOx (15 nM s⁻¹ for 1 nM GOx), the H₂O₂ concentration converges to a constant value so that the actual reaction velocity of HRP is equal to 15 nM s⁻¹."

8) *Why are the activities of GOx and HRP reported in the legend of Fig 5 lower than the values reported in Fig 4?*

Response: We stored the purified GOx-HRP conjugate in 4 °C for 5 days before carrying out the catalase competition experiment. The conjugate retained about 80% activity compared with the activity of the fresh conjugate.

9) *Ref 13 of the manuscript is a review paper, only experimental examples should be cited on line 274. Also, it is not clear why HRP is rate limiting in ref. 11. The ratio of GOx:HRP is 1:1, similar to the experiments shown here in Figs 2 and 4 where HRP is not rate limiting. Please clarify/explain.*

Response: We thanks the reviewer to point out the improper citing of ref 13 here. We agree that HRP does not have to be the rate-limiting enzyme. We corrected the sentence as below:

Page 14, line 298-301: "In the previous publications,^{10,11,15} if the HRP is the limiting enzyme, the lower local pH will significantly enhance the cascade activity; otherwise, the GOx is the limiting enzyme, then the combination of the increased activity of GOx and the shortened transient time by the much more active HRP will also result in enhanced cascade activity."

10) *Ref. 21 of the manuscript is discussed at length with respect to increases in k_{cat} in DNA nanostructure environments. Are the other, non-HRP, enzymes studied in ref. 21 also pH dependent?*

Response: To our knowledge, pH dependence is one of the most common properties of enzymes. Regarding the other enzymes used in Ref. 21, the activity of malic dehydrogenase (*Biochemistry*, 1986, 25, 3752–3759), lactic dehydrogenase (*Biochem J.*, 1982, 203, 393–400) and glucose 6-phosphate dehydrogenase (*Proc. Natl. Acad. Sci. U.S.A.*, 1966, 56, 1543–2547) are all pH dependent. However, the kinetics of these enzymes are controlled by the NAD^+/NADH ratio. In order to focus our discussion on the GOx-HRP cascade, we do not mention other enzymes in the manuscript.

11) Some discussion should be added to explain how the results of Fig 6 can account for the change in GOx-HRP cascade enhancement with changes in interenzyme distance described in ref. 11 of the manuscript. If the DNA nanostructure alters the local pH, thus increasing the apparent kinetics of HRP then why is there an observed distance-dependent effect?

Response: Ref. 11 provides an example of distance-dependent effect on the overall activity. However, the activity of the assembled GOx-DNA and HRP-DNA nano-complexes are not known. For the distance-dependent effect in ref 11, it is more likely dependent on the total amount of DNA origami occupied by the enzymes, rather than the co-assembled GOx-HRP-DNA structures. The steric effect would make more enzymes separately anchor on the DNA, resulting in an increased fraction of DNA-influenced enzyme. We carefully checked the time-course throughput in Figure 2A, ref 11: the activities of GOx-HRP-DNA pairs with distance of 20 nm, 45 nm, and 65 nm show almost the same activity, since they all have high co-assembly yields. The surprising higher activity occurs when the distance is fixed at 10 nm, when the co-assembly yield is only 45% (the yield was determined from AFM images, but the actual co-assembly yield in the solution may be much lower), indicating more GOx or HRP molecules are separately anchored on the DNA origami. It is possible that the activity of the co-assembled enzyme pairs is lower than that of the individually assembled enzyme-DNA nanocomplex because: 1) the close neighbor enzyme may partially block the electrostatic interaction; 2) the crowded placement may limit the swinging range of enzyme thus reducing the residence time in the lower pH area. As a consequence, the lower the co-assembly yield, the higher the observed activity would be. Figure 2B in ref 11 is thus not conclusive since the activity enhancement may not from the closer placement.

In addition, as an indirect evidence for the increased activity by DNA nanostructures, we would like to refer to a result presented in Xin et al's publication (*Small*, 2013, 9, 3088-3091), Figure S5. As we discussed in our paper, if the activity of HRP is much greater than that of GOx, no obvious transient state can be found. In Figure S5A, a clear transient curve occurs in the case of GOx-DNA machine with free HRP, while in Figure S5B no transient state was found in the case of HRP-DNA machine with free GOx, indicating that at least the activity of HRP-DNA machine has increased.

12) The number of technical and biological replicates should be clearly stated in each figure legend.

Response: Thanks, it has been addressed.

13) Nice paper.

Response: Thanks!

Reviewer #2 (Remarks to the Author):

The authors report kinetic measurements and simulation of the GOX-HRP cascade reaction, which is the most prominent example of synthetic distance-dependent enzyme networks studied so far. Owing to the high relevance of the GOX-HRP system for motivating research activities in nanobiotechnology (which is not limited to DNA-arranged enzymes), the key finding that proximity does not contribute to activity enhancement would, in principle, qualify the paper for publication in *Nature Communication*. However, the manuscript appears to be not clearly focused and has also technical flaws.

1. The lack of clear focus is evident from the introductory part (p3, line 47 ff) which aims to justify the relevance of the present work. The authors argue that "differences in the reported activity enhancements (from 1.5 to larger 15-fold)" would justify their work. In fact, all cited examples used different assembly strategies, and thus, it is not too surprising that differences in activity enhancements occur. While this referee strongly supports the notion that postulated mechanisms should lead to quantitatively comparable results, it is only fair to note that different assembly systems would of course lead to different extents of an effect.

Differences in the reported activity enhancements might also stem from insufficient characterization or purification of the derivative enzymes and complexes; enzyme modification often leads to changed activities and remaining free enzymes can distort activity measurements (for a discussion, see Timm, *Angewandte* 2015, 6745) - While this reviewer agrees that necessary controls were not always conducted in previous reports on enzyme cascades, it seems important to note that some of the "1.5 to larger 15-fold" differences could easily be explained by technical imprecisions.

Response: We thank the reviewer for this constructive discussion. This work is aimed to clarify the proximity effect on the activity enhancement of the enzyme cascade. We do agree that the enzyme modification will influence the enzymatic activity. However, in Fu et al.'s paper (*JACS*, 2012, 134, 5516-5519), the oligonucleotides modified enzyme pair retained 80% activity, and in Xin et al.'s paper (*Small*, 2013, 9, 3088-3091) the modified enzyme pair also showed a little bit lower activity than the free enzyme pair, but these small changes have no significant effect on the activity enhancement. The remaining argument is that both of them did not give the individual activity (glucose oxidation rate and H₂O₂ consumption rate) of assembled GOx-HRP-DNA nanostructures.

In Fu et al.'s paper, DNA origami is used to assemble the GOx-HRP pair with controlled distance from 10 nm to 65 nm. In Xin et al.'s paper, a DNA nanostructure is used to anchor the GOx-HRP pair and control the distance change between 6 nm and 18 nm. If the main reason for the activity enhancement is the proximity effect, then a similar gain factor should be observed. However, in Fu et al.'s paper, a 15 times higher activity was claimed compared with the free enzyme control, while in Xin et al.'s paper, the activity only increased about 1.5 fold. Since a similar strategy was used, we believe that this is a fair comparison. We added a discussion in the revised manuscript:

Page 3, line 49-52: "For instance, Fu *et al.*¹¹ co-assembled the GOx-HRP pair on DNA origami and observed 15 times higher activity when the distance between two enzymes is 10 nm. In contrast, Xin *et al.*¹⁷ placed the GOx and HRP even closer using a similar strategy but found

only a 1.5-fold activity enhancement."

2. *The authors argue that several synthetic multi-enzymes did not at all reveal enhanced activity, citing refs 16, 17. This is a bit misleading because ref 17, for instance, concerned entirely different enzymes (in addition to a completely different assembly method). It is thus unclear whether the authors want to address just the HRP-GOX cascade or whether the paper is meant as a general assessment of proximity effects in multi-enzymes? In the latter case, other papers which reported proximity-enhanced activities of DNA-linked enzyme cascades (e.g., Niemeyer, *ChemBiochem* 2002, 242 or Timm, *Angewandte* 2015, 6745) should also be cited.*

Response: We thank the reviewer for providing these key references, which have been cited in our revised version. We aimed to provide a quantitative evaluation of the proximity effect on the enzyme cascade using the GOx and HRP model system.

Page 3, line 41-43 : "Several other enzyme cascades also showed enhanced activity when they are co-assembled with DNA nanostructures.^{12,15,16"}

3. *Furthermore, the authors conclude a contradiction from their literature review that the reported activity enhancements appear to exist indefinitely "by citing refs.11,15,21. Having checked the three papers, this referee does not understand this argumentation.*

Response: The word "indefinitely" is a bit imprecise. We have revised this sentence as below:

Page 3, line 64-67: " Proximity in itself should not influence the maximal reaction rate of either enzyme, however the reported activity enhancements appear to persist over the entire observation time which would indicate that the limiting maximal reaction rate has increased."

4. *The data shown in Figures 1 - 3 appear to be sound. Michaelis-Menten parameters of HRP have certainly been measured in several other studies before – could the authors please compare their data with previously reported ones for HRP and GOX.*

Response: We provided the nonlinear fitting of Michaelis-Menten kinetics for GOx and found that the k_{cat} is 250 s^{-1} in **Supplementary Figure 1**. In Fu et al. (*JACS*, 2012, 134, 5516-5519), the k_{cat} was 300 s^{-1} , very close to our observations. The activity parameters of HRP can be found in the latest paper from the Yan group (*Nat. Commun.* 2016, 7), where the k_{cat} and K_m of HRP are almost as same as in our paper (we measured a k_{cat} of 33 s^{-1} and K_m of $2.5 \mu\text{M}$, in comparison, they obtained a k_{cat} of 32 s^{-1} and K_m of $2.3 \mu\text{M}$).

5. *The experimental details for determination of the Michaelis-Menten parameters of GOX are too scarce; please describe precisely which instrumentation was used.*

Response: Thanks, we have introduced the method in detail. And the Michaelis-Menten fitting has been added as **Supplementary Figure 1**.

6. *Can the authors exclude that the kinetic data are inaccurate due to the huge amounts of impurities (>80 %!) evident from Fig. S7 ? What is the nature of these impurities? Were the data shown in Figures 1-3 generated with purified or non-purified HRP?*

Response: We purchased the HRP from Sigma Aldrich and realized the problem of purity after checking the SDS-PAGE gel. We then further purified the HRP by size exclusion chromatography. The purified HRP has a RZ value of 2.5 (RZ value is the absorbance ratio A_{403}/A_{275} and a

measure of the hemin content of the peroxidase), indicating acceptable purity. We carried out the experiment with the purified HRP.

7. *Fig. 5: Which cascade was studied here? Isolated enzymes or the chemical conjugates? I would expect that both systems should be compared.*

Response: The experiment of Figure 5 was carried out with the conjugated enzyme. The free enzyme comparison has been added in the revised Supplementary Information, Figure S15. We have clarified the Figure caption 5b: “The suppression of ABTS⁺ production from the GOx-HRP conjugate by addition of catalase, here the reaction velocity of GOx is 11.8 nM/s, and V_{max} of HRP is 25.3 nM/s. For the free GOx-HRP experiment see Supplementary Fig. 15.”

8. *Fig. S10: How were the spectra normalized? This kind of representation is highly prone to artefacts when different amounts/concentrations of proteins are used. Hence, protein needs to be thoroughly quantified before – how has this been done here? A control should be included in which equal amounts of pure GOX and HRP are mixed with each other.*

Response: We did not normalize the spectra. The fraction of conjugated enzyme from the chromatography was concentrated by ultrafiltration and the concentration was adjusted by its absorbance at 280 nm and 403 nm. We scanned the spectra at fixed concentrations (1 μM) of HRP, HRP-SMCC, GOx and GOx-SH and conjugate. The spectrum of mixed GOx and HRP has been added in Figure S11 (formerly Figure S10).

9. *Fig. 6 and the related discussion: The fact that origami samples are typically conducted in TEMg buffer (containing 12.5 mM MgCl₂) seems to be neglected. Divalent Mg cations should significantly alter the surface charge of polyelectrolytes and also affect the changes in pH. The authors used "10 mM salt" (sodium chloride?) but magnesium ions have significantly different solvation properties. In addition to taking this into account for their simulations, the authors should also experimentally verify that the activity of their enzymes is not affected by the TEMg buffer (20 mM Tris, 2 mM EDTA, 12.5 mM MgCl₂, pH = 7.6)*

Response: We determined the activity of GOx, HRP and conjugate in PBS and TEMg buffer respectively, as shown in Supplementary Figure 16, the activities of GOx and HRP do not significantly change in TEMg buffer. We have stated that in the main text in **Page 12 , line 228 :** "We checked that the activity of GOx and HRP is not affected by the buffer used for DNA origami (Supplementary Fig. 16)"

10. *Fig. 6c: The activity of GOX appears to be much higher than that shown in Fig. 4d. How are these differences explained?*

Response: The pH dependence of GOx was measured in 100 mM glucose, while the activity in the former paragraph are determined in 1 mM glucose. As suggested by the reviewer, we have changed Figure 6c showing the activity of GOx in 1 mM glucose.

11. *Based on the author's conclusions, I am still puzzled and, for instance, do not understand how results of Ref 11 can be explained where DNA-coupled GOX and HRP enzymes were compared to each other. Furthermore, the enhancement of enzyme activity through conjugation with DNA was already well documented from previous studies (see Timm, mentioned above; Fruk, Chem. Eur. J.*

2006, 7448; Glettenberg, *Bioconj. Chem.* 2009, 969; Rudiuk, *Angewandte* 2012, 12694). Hence, the novelty of findings might be disputable and the question remains what we can learn from this paper for future studies? Do the authors expect that proximity does not contribute to activity enhancement of other cascades as well?

Response: The main point in this study is that the permanent activity enhancement of enzyme cascade cannot be explained by the proximity effect or substrate channeling. Our insights are: 1) If the individual activity of each enzyme is fixed, then the final activity should be limited by the slower one. Some of the previous measurements were made during the transient state when reaction has not yet reached the steady state. The only benefit from an improved transport is to shorten the transient time for reaching the limited activity.

2) If others claimed that the overall activity is enhanced by the proximity effect, then more evidence should be provided and other factors should be carefully ruled out. This applies to Ref.11, where the individual activity of the assembled GOx-DNA and HRP-DNA nano-complexes was not measured and it is not clear if there is a steady state.

Although we are the first to suggest that the activity enhancement originates from a lower pH on the DNA surface, the shift of the pH-profile of the enzyme activity by a polyelectrolyte in the enzyme's vicinity is not a new finding in the field of enzyme immobilization. Goldstein *et al.* immobilized trypsin on a water-insoluble polyanionic carrier and found that the pH-activity profile of immobilized trypsin shifted toward more alkaline pH values by 1-2.5 pH units relative to the native form, and this effect was highly dependent on the ionic strength (*Biochemistry*, 1964, 3, 1913-1919). Similar study can be found for chymotrypsin (*Biochemistry*, 1972,11,4072-84) and papsin (*Biochemistry*, 1970, 9, 2322-2334). We have cited these important references in the revision.

For other cascades, to support the claim that proximity caused activity enhancement, two factors should be carefully evaluated and ruled out: 1) Is there a steady state and has it been reached; 2) which enzyme is the rate-limiting one and did the individual activity of this enzyme change.

Reviewer #3 (Remarks to the Author):

Comments for Manuscript:

Proximity Does Not Contribute to Activity Enhancement in the Glucose Oxidase-Horseradish Peroxidase Cascade

Yifei Zhang, Stanislav Tsitkov, Henry Hess

This manuscript presents a detailed look into explaining the mechanism of activity improvement of the glucose oxidase (GOx) and horseradish peroxidase (HRP) enzyme cascade on DNA scaffolds. Importantly, this work provides some direct investigation into the cause of improved activity due to enzyme localization to a DNA scaffold. Up until now, these improvements have been attributed to a substrate channelling or proximity effect due to the co-localization of the two enzymes onto the DNA scaffold. However, through a series of computational and enzymatic experiments examining the effects of varying the limiting enzyme, adding a competitive enzyme and fusing the enzymes via a different linker, the authors show that the increased throughput observed in previously published reports on the GOx-HRP cascade on DNA scaffold is not due to a proximity effect, but rather is a

result of the local environment of low pH created by the negatively charged DNA backbone.

1. Page 2, Line 21: The authors appear to generalize the rate enhancements of enzymes on DNA scaffolds as a result of the local lower pH, but do not appear to provide sufficient evidence that this applies to enzymes other than the glucose oxidase and horseradish peroxidase cascade. While GOx and HRP may be the most common enzymes to use in demonstrating a cascade on a DNA scaffold, it is not the only enzyme set used. It may be more appropriate to conclude that the presented results ‘overturn’ the general proximity effect for the GOx HRP pair on DNA scaffolds, and also ‘challenges’ the accuracy of this in other cascades. This effect may indeed apply to other enzymatic cascades tethered to DNA, however without explicit examination this cannot yet be stated. A similar comment can be applied to the usage on page 3, line 69.

Response: We thank the reviewer for this kind suggestion. We have replaced the ‘overturn’ by ‘challenge’ in both abstract and main text.

2. Page 3, Line 51: Please clarify how a transient effect can be extended if it only has an influence until the intermediate reaches a significant concentration. Also please clarify how aggregated enzymes, attractive interactions between intermediate and scaffold, as well as competing reactions can all provide the same effect, when the latter is later shown to be deleterious to reaction throughput.

Page 11: Catalase, a competing reaction to GOx and HRP, is described as suppressing formation of ABTS⁺ in both the conjugated and free enzyme forms, but on page 12, line 212 the authors reference papers by Idan and Hess that have suggested that competing reactions can lead to throughput enhancements. Please clarify how both effects are possible.

Response: These conclusions were discussed in our previous paper (*ACS Nano*, 2013, 7, 8658 – 8665) by Monte Carlo simulation. The key to extending the transient effect is postponing the establishment of the steady state. The aggregation provides more target enzymes in the vicinity of the first enzyme and also decreases the possibility of intermediate leaking. The attractive interaction between intermediate and scaffold will increase the local concentration of intermediate and prevent its leaking to the bulk solution. For the competing reaction, the accumulation of the intermediate in the bulk solution will slow down in the presence of competing reaction. However, the addition of a competing reaction depresses the overall activity since it decreases the concentration of intermediate. To avoid the confusion, we deleted this statement, and added more discussion in the main text.

Page 3 , line 59-63:"In brief, the aggregation provides more target enzymes in the vicinity of the first enzyme, and the attractive interaction can increase the local concentration of intermediate and prevent its leaking to the bulk solution. These effects enhance the production rate of the second enzyme, and also extend the channeling time due to the delayed establishment of the steady state."

4. Page 13, Figure 6c: If only the activity of HRP increases at lower pH (free glucose oxidase sharply loses activity below pH 5.5), how does that fit with the proposed effect of the reduced pH at the DNA backbone being responsible for activity improvements? Also, to be more comparable to the other data presented here and to that presented in the referenced articles, the pH-activity profile for GOx may be more informative if tested at 1 mM glucose. This will demonstrate the effect of pH on the conditions used with the DNA scaffold, and also be more representative to the

proposed effect.

Response: We thank the reviewer for this suggestion. We have replaced Figure 6c with the pH profile determined in 1 mM glucose. GOx also showed an increased activity in more acidic condition but not as significant as HRP. The activity enhancement can come from two reasons even if only the activity of HRP increased. 1) if the HRP is the limiting enzyme, then the overall activity will be significantly enhanced. 2) if the GOx is the limiting enzyme, then the combination of increased activity of GOx and the shortened transient time by the much more active HRP will also result in enhanced cascade activity. We also added more discussion in the main text (**Page 14, line 298-301**).

5. An experiment I would like to see included in the manuscript is a pH activity (or cascade throughput rate) profile of the GOx-HRP conjugate. This could likely most directly show the pH dependence of the cascade and also provide the appropriate enzyme proximity and stoichiometry compared to the cascade on a DNA scaffold.

Response: We carried out the reaction at different pH values with the GOx-HRP conjugate and have added the results in the Supplementary Information, Figure S16. As expected, the final activities in different pH are limited by the GOx, in accordance with the pH-activity profile shown in Figure 6c. However, HRP may be the limiting enzyme for the cascades in the literature if the activity of the GOx is higher than our GOx activity. For instance, in Fu *et al.*'s paper (*JACS*, 2012, 134, 5516-5519, Figure S11 A), the activity of 1 nM GOx in 1 mM glucose is about 70 nM/s.

6. Minor notes:

Page 1, Line 19: '...production by the GOx-HRP conjugate,...'

Page 3, Line 68: '...where the enzyme is located.'

Page 5, Line 100: '...concentration as a function of time for different ratios of HRP to GOx...'

Page 15, Line 297: 'sulphonic'

Response: Thanks, these mistakes have been corrected.

7. This work provides a nice, detailed study and presents some mechanistic insight to the cause of activity improvements in cascades tethered to DNA scaffolds. With the minor additional experiments and text clarifications, I would recommend this manuscript for publication.

Response: Thanks!

Reviewers' comments:

Reviewer #1 (Remarks to the Author):

Thank you for the edits to the manuscript and answers to my concerns. I have no additional comments.

Reviewer #2 (Remarks to the Author):

While the quality of the manuscript has been significantly improved upon revision, several weak points persist which regard the clear focus of the work and (larger) technical flaws (see below).

Page 14, line 298-301: "In the previous publications,10,11,15 if the HRP" ... Wrong citation: Ref 15 does not deal with HRP – did the authors mean Ref 14?.

The problem remains that several points in the discussion are imprecise: If the authors focus on HRP / GOX, the sentence in Line 53 („Secondly, in several reports the colocalization of enzymes did not result in substrate channeling.18,19“) should be removed from this passage because these papers do not concern HRP / GOX

The author's response to point 4 (referee 2, comparison of kinetic data with previously reported data) should be included into the manuscript or at least SI.

Wrt point 5: The experimental details for determination of the Michaelis-Menten parameters of GOX are too scarce; please describe precisely which instrumentation was used. Response: Thanks, we have introduced the method in detail. And the Michaelis-Menten fitting has been added as Supplementary Figure 1. Although stated otherwise in the rebuttal letter, no more details have been added.

Wrt point 6: Can the authors exclude that the kinetic data are inaccurate due to the huge amounts of impurities (>80 % !) evident from Fig. S7 ? What is the nature of these impurities? Were the data shown in Figures 1-3 generated with purified or non-purified HRP? Response: We purchased the HRP from Sigma Aldrich and realized the problem of purity after checking the SDS-PAGE gel. We then further purified the HRP by size exclusion chromatography. The purified HRP has a RZ value of 2.5 (RZ value is the absorbance ratio A403/A275 and a measure of the hemin content of the peroxidase), indicating acceptable purity. We carried out the experiment with the purified HRP.

The referee's questions were completely ignored. Instead, details on the spectroscopic characterization of a purified enzyme batch are given.

Wrt point 9: Fig. 6 and the related discussion: The fact that origami samples are typically conducted in TEMg buffer (containing 12.5 mM MgCl₂) seems to be neglected. Divalent Mg cations should significantly alter the surface charge of polyelectrolytes and also affect the changes in pH. The authors used "10 mM salt" (sodium chloride?) but magnesium ions have significantly different solvation properties. In addition to taking this into account for their simulations, the authors should also experimentally verify that the activity of their enzymes is not affected by the TEMg buffer (20 mM Tris, 2 mM EDTA, 12.5 mM MgCl₂, pH = 7.6)

Response: We determined the activity of GOx, HRP and conjugate in PBS and TEMg buffer respectively, as shown in Supplementary Figure 16, the activities of GOx and HRP do not significantly change in TEMg buffer. We have stated that in the main text in Page 12, line 228 : "We checked that the activity of GOx and HRP is not affected by the buffer used for DNA origami (Supplementary Fig. 16)"

While the experimental testing rules out concerns of buffer artefacts, the referee's point is not

properly addressed: The lengthy description of DNA Origami Potential Profile modelling and estimation of local pH (main text and SI) is still ignoring the effect of divalent cations.
Minor: check legend of Fig. S16 for errors: „ enzymatic activity may different in“ and „ which usually used to“

Wrt point 11: The authors state: „For other cascades, to support the claim that proximity caused activity enhancement, two factors should be carefully evaluated and ruled out: 1) Is there a steady state and has it been reached; 2) which enzyme is the rate-limiting one and did the individual activity of this enzyme change.“ Please include this clear statement in the main text!

Reviewer #3 (Remarks to the Author):

I believe the authors have sufficiently addressed all comments regarding experimental additions from the referees, and all data and discussion of results appear to be well reasoned. The only small suggestions that remain are to clarify some background detail in the introduction and to adjust a reference to previous related work to better suit the present study, with each point expanded below. With these modification, I still recommend this manuscript for publication.

Page 3, Line 59-63: If the steady state is when the cascade is only limited by the maximal rate of the slower enzyme (the author's apparent definition from line 145), why is delaying this (i.e. extending the initial transient stage) a benefit to the system? How would this steady state delayed by aggregating the enzymes or by introduction of attractive enzymes when each work to increase the activity of enzyme two and increasing the activity of enzyme two is shown to decrease the transient time as modelled in Figure 2 by increased HRP? Is the same term, 'steady state', used with a different meaning in different portions of the text, such a relation to 'channelling time' as the time when substrate flux from enzyme 1 is equal to flux from the bulk solution? Please clarify.

Page 13, Line 263: Please add a brief analysis on the increase or decrease in activity of the enzymes anchored to polyanionic and/or polycationic carriers in the references mentioned. This seems to be more relevant to the present example of increased activity on the DNA scaffold rather than the discussion of a shift in pH profile. A change in the pH-activity profile can, in part, be due to buffering against the altered local pH at the surface of the ionic carrier, as demonstrated by a shift toward alkaline pH for a polyanionic carrier (lower local pH) and a shift toward acidic pH for polycationic carrier (higher local pH).

Response to Reviewer's Comments

We appreciate the time and effort expended by the editor and referees in reviewing our manuscript. We have addressed all issues and responded to their useful suggestions and comments. The details can be found below and are also highlighted in the revised manuscript.

Reviewers' comments:

Reviewer #1 (Remarks to the Author):

Thank you for the edits to the manuscript and answers to my concerns. I have no additional comments.

Response: Thank you!

Reviewer #2 (Remarks to the Author):

While the quality of the manuscript has been significantly improved upon revision, several weak points persist which regard the clear focus of the work and (larger) technical flaws (see below).

1) Page 14, line 298-301: "In the previous publications,^{10,11,15} if the HRP " ... Wrong citation: Ref 15 does not deal with HRP – did the authors mean Ref 14?.

Response: We thank the reviewer for pointing out this error. It should be Ref 17 in the new version. We have corrected it in the revised manuscript.

2) The problem remains that several points in the discussion are imprecise: If the authors focus on HRP/GOX, the sentence in Line 53 ("Secondly, in several reports the colocalization of enzymes did not result in substrate channeling.^{18,19}") should be removed from this passage because these papers do not concern HRP/GOX.

Response: Thank you, we have removed this sentence.

3) The author's response to point 4 (referee 2, comparison of kinetic data with previously reported data) should be included into the manuscript or at least SI.

Response: Thank you, we have added the comparison in the revised Supplementary Information as suggested:

"The k_{cat} of GOx and K_m were determined as 250 s^{-1} and 15 mM , respectively. The turnover number is close to the previous study by Fu *et al.*, where the k_{cat} was 310 s^{-1} , but their K_m is 3.5 mM .¹ Therefore the apparent activity of our GOx in 1 mM is 15 nM s^{-1} , but their GOx is 70 nM s^{-1} . For the kinetics of HRP, we measured a k_{cat} of 32.7 s^{-1} and K_m of $2.5 \text{ }\mu\text{M}$. In comparison, Fu *et al.* obtained a k_{cat} of 32 s^{-1} and K_m of $2.3 \text{ }\mu\text{M}$,² where the parameters are almost the same. These comparisons indicate that the rate limiting enzyme is GOx in this study, but may be HRP in previous studies by Fu *et al.*"

4) Wrt point 5: The experimental details for determination of the Michaelis-Menten parameters of

GOX are too scarce; please describe precisely which instrumentation was used. Response: Thanks, we have introduced the method in detail. And the Michaelis-Menten fitting has been added as Supplementary Figure 1. Although stated otherwise in the rebuttal letter, no more details have been added.

Response: We did not intend to ignore the comment, but we placed the added information in the main text where the reader may not look for it. We have rearranged it, and the experimental procedure for Supplementary Figure 1 is now directly in front of it:

"The activity assays were carried out on a UV-Visible spectrophotometer (Evolution 201, Thermo Scientific, US). Briefly, 20 μ L of 50 nM GOx was added to 980 μ L of substrate solution (in PBS buffer, pH 7.4) to initiate the reaction. The substrate buffer contains 2 mM ABTS, 20 nM HRP and different concentrations of glucose (1 mM to 90 mM). The increase in absorbance at 415 nm was continuously recorded for 2 min and the progress curve was fitted with a linear fit to calculate the GOx activity."

5) Wrt point 6: Can the authors exclude that the kinetic data are inaccurate due to the huge amounts of impurities (>80 % !) evident from Fig. S7 ? What is the nature of these impurities? Were the data shown in Figures 1-3 generated with purified or non-purified HRP? Response: We purchased the HRP from Sigma Aldrich and realized the problem of purity after checking the SDS-PAGE gel. We then further purified the HRP by size exclusion chromatography. The purified HRP has a RZ value of 2.5 (RZ value is the absorbance ratio A403/A275 and a measure of the hemin content of the peroxidase), indicating acceptable purity. We carried out the experiment with the purified HRP.

The referee's questions were completely ignored. Instead, details on the spectroscopic characterization of a purified enzyme batch are given.

Response: Maybe this is a miscommunication, but we aimed to address the reviewer's comment by stating that the experiments were carried out with the purified HRP and we tried to support that the purified HRP is not contaminated anymore by spectroscopic characterization. The impurities should be other proteins since they contribute to the absorbance at 280 nm. We know that the impurities do not affect the activity of HRP, due to the evidence shown in Figure S8 (b). Although we do not exactly know what the impurities are, we have removed them in our experiments, including the data shown in Figures 1-3. Thus we believe that the kinetic data for the purified enzyme are accurately reflecting the enzyme properties.

The methods section now tries to clearly state this: "For all experiments, the purchased HRP was first purified by size exclusion chromatography to remove the impurities (Supplementary Fig. 7 and 8)."

6) Wrt point 9: Fig. 6 and the related discussion: The fact that origami samples are typically conducted in TEMg buffer (containing 12.5 mM MgCl₂) seems to be neglected. Divalent Mg cations should significantly alter the surface charge of polyelectrolytes and also affect the changes in pH. The authors used "10 mM salt" (sodium chloride?) but magnesium ions have significantly different solvation properties. In addition to taking this into account for their simulations, the authors should also experimentally verify that the activity of their enzymes is not affected by the TEMg buffer (20 mM Tris, 2 mM EDTA, 12.5 mM MgCl₂, pH = 7.6)

Response: We determined the activity of GOx, HRP and conjugate in PBS and TEMg buffer

respectively, as shown in Supplementary Figure 16, the activities of GOx and HRP do not significantly change in TEMg buffer. We have stated that in the main text in Page 12 , line 228 :"
We checked that the activity of GOx and HRP is not affected by the buffer used for DNA origami (Supplementary Fig. 16)"

While the experimental testing rules out concerns of buffer artefacts, the referee's point is not properly addressed: The lengthy description of DNA Origami Potential Profile modelling and estimation of local pH (main text and SI) is still ignoring the effect of divalent cations.

Response: We do agree that the divalent ions can affect the surface charges of DNA origami. As we clearly state, the modeling of the pH profile omits several effects of known importance, and is merely a rough approximation. Moreover, people do use TEMg buffer to dissolve DNA origami samples but do not use it for the activity assay. We have checked the previous publications and found that the concentration of Mg^{2+} in the assay buffer is not as high as 12.5 mM, in fact some of them did not use Mg^{2+} in the assay buffer. For instance, Fu *et al.* (JACS, 2012, 134, 5516-5519) used TBS buffer (tris buffered saline, pH 7.5) with 1 mM $MgCl_2$ for the GOx-HRP activity assay. Müller and Niemeyer (Biochem. Biophys. Res. Commun., 2008, 62-67) used Kpi300 buffer (50 mM Potassium phosphate, 300 mM NaCl, pH 7.4) to measure the activity. Wilner *et al.* (Nat. Nanotechnol., 4, 249-254) performed all their assays in a 10 mM phosphate buffer solution. These publications all claimed the proximity induced activity enhancements. Considering that the concentrations of DNA-bounded enzyme used in these activity assays were on the order of nM, the Mg^{2+} introduced to the buffer by adding the DNA structures should be negligible. Therefore, there is no evidence that divalent cations play a critical role and integrating them in the model would only extend the description of the model.

For the "10 mM salt" in the modeling part, it means 10 mM of monovalent salt. We have clarified it in **Page 14, line 281**.

7) Minor: check legend of Fig. S16 for errors: "enzymatic activity may differ in" and "which usually used to"

Response: Thanks. We have addressed these errors.

In Supplementary Information, Figure S16: "A concern is that the enzymatic activity may be different in phosphate buffer and TEMg buffer (20 mM Tris, 2 mM EDTA, 12.5 mM $MgCl_2$, pH=7.5), which is usually used to dissolve DNA origami samples."

8) Wrt point 11: The authors state: "For other cascades, to support the claim that proximity caused activity enhancement, two factors should be carefully evaluated and ruled out: 1) Is there a steady state and has it been reached; 2) which enzyme is the rate-limiting one and did the individual activity of this enzyme change." Please include this clear statement in the main text!

Response: We thank the reviewer for this kind suggestion. We have added this in the revised version in **Page 15, line 319-323**:

"Based on these insights, to support the claim that proximity caused activity enhancement in other cascades, two factors should be carefully evaluated and ruled out: 1) Is there a steady state and has it been reached; 2) which enzyme is the rate-limiting one and did the individual activity of this enzyme change."

Reviewer #3 (Remarks to the Author):

I believe the authors have sufficiently addressed all comments regarding experimental additions from the referees, and all data and discussion of results appear to be well reasoned. The only small suggestions that remain are to clarify some background detail in the introduction and to adjust a reference to previous related work to better suit the present study, with each point expanded below. With these modification, I still recommend this manuscript for publication.

1) Page 3, Line 59-63: If the steady state is when the cascade is only limited by the maximal rate of the slower enzyme (the author's apparent definition from line 145), why is delaying this (i.e. extending the initial transient stage) a benefit to the system? How would this steady state delayed by aggregating the enzymes or by introduction of attractive enzymes when each work to increase the activity of enzyme two and increasing the activity of enzyme two is shown to decrease the transient time as modeled in Figure 2 by increased HRP? Is the same term, 'steady state', used with a different meaning in different portions of the text, such a relation to 'channelling time' as the time when substrate flux from enzyme 1 is equal to flux from the bulk solution? Please clarify.

Response: The "short-lived (ms) effect" of facilitated transport due to proximity does not refer to what is commonly named "transient stage". During the transient stage (on the timescale of seconds to minutes), the intermediate substrate concentration builds up in the solution. The normally short-lived effect of facilitated transport due to proximity can be extended to seconds to minutes in special situations (Ref. 18), where losses to alternative reactions or extremely low enzyme concentrations prevent the rapid accumulation of intermediate substrate in the solution. In these situations, channeling is increasing the rate of product formation over the value obtained without channeling at the same time point. The extent of the losses (e.g. due to alternative reactions in the intracellular environment) is central to the cascade kinetics. However, most in vitro experiments do not correspond to these special situations.

In the new version, we consistently use "steady state" for the state when the product formation rate is constant and limited by the activity of the slower enzyme.

To clarify the statement on Page 3, Line 59-63, we revised it as below:

Page 3, line 52: "Secondly, a theoretical analysis of the GOx-HRP cascade by Idan and Hess^{18,19} challenged the proximity channeling effect. They estimated the time scale when the intermediate flux from the upstream enzyme is equal to the flux from the bulk solution and pointed out..."

2) Page 13, Line 263: Please add a brief analysis on the increase or decrease in activity of the enzymes anchored to polyanionic and/or polycationic carriers in the references mentioned. This seems to be more relevant to the present example of increased activity on the DNA scaffold rather than the discussion of a shift in pH profile. A change in the pH-activity profile can, in part, be due to buffering against the altered local pH at the surface of the ionic carrier, as demonstrated by a shift toward alkaline pH for a polyanionic carrier (lower local pH) and a shift toward acidic pH for polycationic carrier (higher local pH).

Response: We thank the reviewer for this constructive suggestion. We have added a brief discussion as following:

Page 14, line 298: "Also, according to the studies by Goldstein,³⁶⁻³⁸ if the GOx and HRP are anchored on a polyanionic carrier that can decrease the local pH by 2 units (from 7.5 to 5.5), then their activity will increase by 1.2-fold and 4-fold respectively, leading to a enhanced overall activity."

REVIEWERS' COMMENTS:

Reviewer #2 (Remarks to the Author):

The answers to my questions and the changes to the manuscript address the majority of my concerns. I have one final point:

Point 5: There are still no experimental details how the data shown in Figure 1b (red graph) were produced, i.e. how was the oxygen consumption measured. This is not trivial and the authors should explain which specific instrumentation and reaction conditions were used.

Response to Reviewer

REVIEWERS' COMMENTS:

Reviewer #2 (Remarks to the Author):

The answers to my questions and the changes to the manuscript address the majority of my concerns. I have one final point:

Point 5: There are still no experimental details how the data shown in Figure 1b (red graph) were produced, i.e. how was the oxygen consumption measured. This is not trivial and the authors should explain which specific instrumentation and reaction conditions were used.

Response: We thank the reviewer for his attention to our manuscript. We described the measurement of oxygen consumption in the activity assay section. In this revision, we provided more details:

"The consumption of oxygen by GOx was measured by a dissolved oxygen meter (Mettler Toledo, FiveGoTM) in PBS buffer containing 1 mM D-glucose in a sealed vessel. We placed the probe of the dissolved oxygen meter in a glass vessel fully filled with 3 mL substrate buffer. As soon as 10 μ L of 300 nM GOx was added to the solution (final concentration is 1 nM), the vessel is immediately sealed with a piece of plastic paraffin film. The oxygen concentration was recorded for 1 h under magnetic stirring."